# Along-strike variation of volcanic addition controlling post breakup sedimentary infill: Pelotas margin, Austral South Atlantic

Marlise C. Cassel[1], Nick Kusznir[2], Gianreto Manatschal[3], Daniel Sauter[3]

[1]Geological Institute / Organic Biochemistry in Geo-Systems Institute, RWTH Aachen University, Aachen, 52062, Germany
[2]School of Environmental Sciences, Liverpool University, Liverpool L69 3GP, UK
[3]Université de Strasbourg, CNRS, ITES UMR 7063, Strasbourg 67084, France

*Correspondence to*: Marlise C. Cassel (marlise.cassel@gmail.com)

**Abstract.** We investigate, using observations from seismic reflection data, the lateral variability of breakup extrusive magmatic addition along-strike of the Pelotas segment of the Austral South Atlantic rifted margin and its control on post-rift accommodation space and sediment deposition. Our analysis of regional seismic reflection profiles shows that magmatic addition on the Pelotas margin varies substantially along strike from extremely magma-rich to magma-normal within a distance of ~300 km. Using 2D flexural backstripping we determine the post-rift accommodation space above top volcanics. In the north, where SDRs (volcanic seaward dipping reflector) are thickest, the Torres High shows SDRs up to ~ 20 km thick and post-breakup water-loaded accommodation space is much less than in the south where magmatic addition is normal and SDRs are thinner. We show that post-breakup accommodation space correlates inversely with SDR thickness, being less for magma-rich margins and more for magma normal/intermediate margins. The Rio Grande Cone, with large sediment thickness, is underlain by small SDR thicknesses allowing large post-breakup accommodation space. A relationship is observed between the amount of volcanic material and the TWTT of first volcanics; first volcanics are observed between 1.2 and 2.2s TWTT for the highly magmatic Torres High profile while, in contrast, for the normally magmatic profiles in the south, first volcanics are observed between 4.2 and 6.5s TWTT. The observed inverse relationship between post-breakup accommodation space and SDR thickness is consistent with predictions by a simple isostatic model of continental lithosphere thinning and magmatic addition melting during breakup. The methodology that we use in this paper provides a new approach for investigating the complex magmatic and sedimentary evolution of rifted continental margins.

## 1 Introduction

Much recent continental margin research has been focussed on either magma-rich margins showing thick sequences of magmatic extrusives or in contrast magma-poor margins showing domains of exhumed mantle between thinned continental crust and new magmatic ocean crust. Sapin et al. (2021) however point out that these represent end-members of rifted margin magmatic type and that a continuous spectrum may exist between them. In this paper we investigate, using observations from seismic reflection data, lateral variability of breakup volcanic addition along-strike of the Pelotas segment of the Austral South

Atlantic rifted margin and its control on post-rift accommodation space and sediment deposition. Breakup along the Austral segment of the S-Atlantic occurred by the propagation of rift systems accompanied by extensive magmatism, resulting in classical volcanic margins characterized by seaward-dipping reflectors (SDRs) (Koopman et al., 2014). While SDRs have been mapped and described in detail through dip sections along the Austral segment at both conjugate margins (Chauvet et al., 2021), much less is known about the along strike evolution of the magmatic system. Stica et al. (2014) and Franke et al. (2007) described the along strike evolution of the magmatic breakup, suggesting that the magmatic system was laterally continuous, with breakup evolution being controlled by the Tristan mantle plume resulting in the Paraná-Etendeka magmatic province (Thompson et al., 2001; Peace et al. 2020). In contrast, a more recent study by Sauter et al. (2023) shows that the magmatic budget along large parts of the Austral segment does not need a hot-mantle booster and that higher magmatic budgets can only be observed north of the Chui-Cape Cross Fracture Zone when approaching the Paraná-Etendeka magmatic province. Sauter et al. (2023) analysed, however, only the magmatic budget recorded in the first oceanic crust. Important remaining questions are: do variations in magmatic addition occur along the Pelotas margin and, if variations do occur, how are they manifest in the margin architecture and how do they control margin accommodation space and depositional history.

The approach we use in our investigation is to restrict our observations from seismic reflection data to those which do not depend on speculative interpretations. As a consequence:

(i) We do not consider the nature of the basement onto which extrusive magmatism is deposited; the identification of whether basement is thinned continental crust, oceanic crust or hybrid is imprecise and ambiguous.

(ii) We focus on the more proximal extrusive magmatism and avoid taking measurements where it transitions into oceanic layer 2.

(iii) We only take measurements for extrusive magmatism and do not consider intrusive magmatism which cannot be reliably observed or quantified.

(iv) We do not consider the formation processes of extrusive magmatism; we focus on measured observations from seismic reflection data.

(v) We preferentially take our measurements from time domain seismic reflection sections which are the primary observational data set. Depth converted seismic sections are model dependent on the seismic velocities used in depth conversion; at magma-rich margins seismic velocities are highly heterogeneous and uncertain (McDermott et al., 2019) and the resultant depth sections are unreliable.

Our study focuses on the Pelotas margin of the Austral South Atlantic located north of the Chui-Cape Cross Fracture Zone and offshore of the southeastern border of the Parana magmatic province (Fig. 1). We use four parallel deep long-offset seismic reflection dip lines that allow us to determine the along strike variation of volcanic thickness, sediment thickness and post-breakup accommodation space. We first examine the relationship between thicknesses of SDRs and sediments in two-way travel time (TWTT). We then use 2D flexural back-stripping of depth converted sections to determine post-rift accommodation space and its relationship with breakup volcanic addition. Our results reveal a direct relationship between the volume of breakup volcanics and post-breakup sedimentary fill along the Pelotas rifted margin.

SDRs with long flow lengths and large thicknesses, which form by extrusive magmatism in a sub-aerial environment, have been extensively studied (Mutter et al., 1982, Planke et al., 2000; McDermott et al., 2019; Harkin et al., 2019). However volcanic SDRs also form in a deep marine environment as voluminous effusive sheet flows (Planke et al., 2000) as shown in Hinz et al. (1999, figure 14), Planke et al. (2000, figure 9) and Sapin et al. (2021, figure 6), the top of these deep marine SDRs often showing perfect distal continuity with the top of oceanic layer 2. In this paper we use the term SDRs to denote the general

observation of volcanic sea-ward dipping reflectors not only applied to those formed in a sub-aerial environment but also to those formed in deep water.

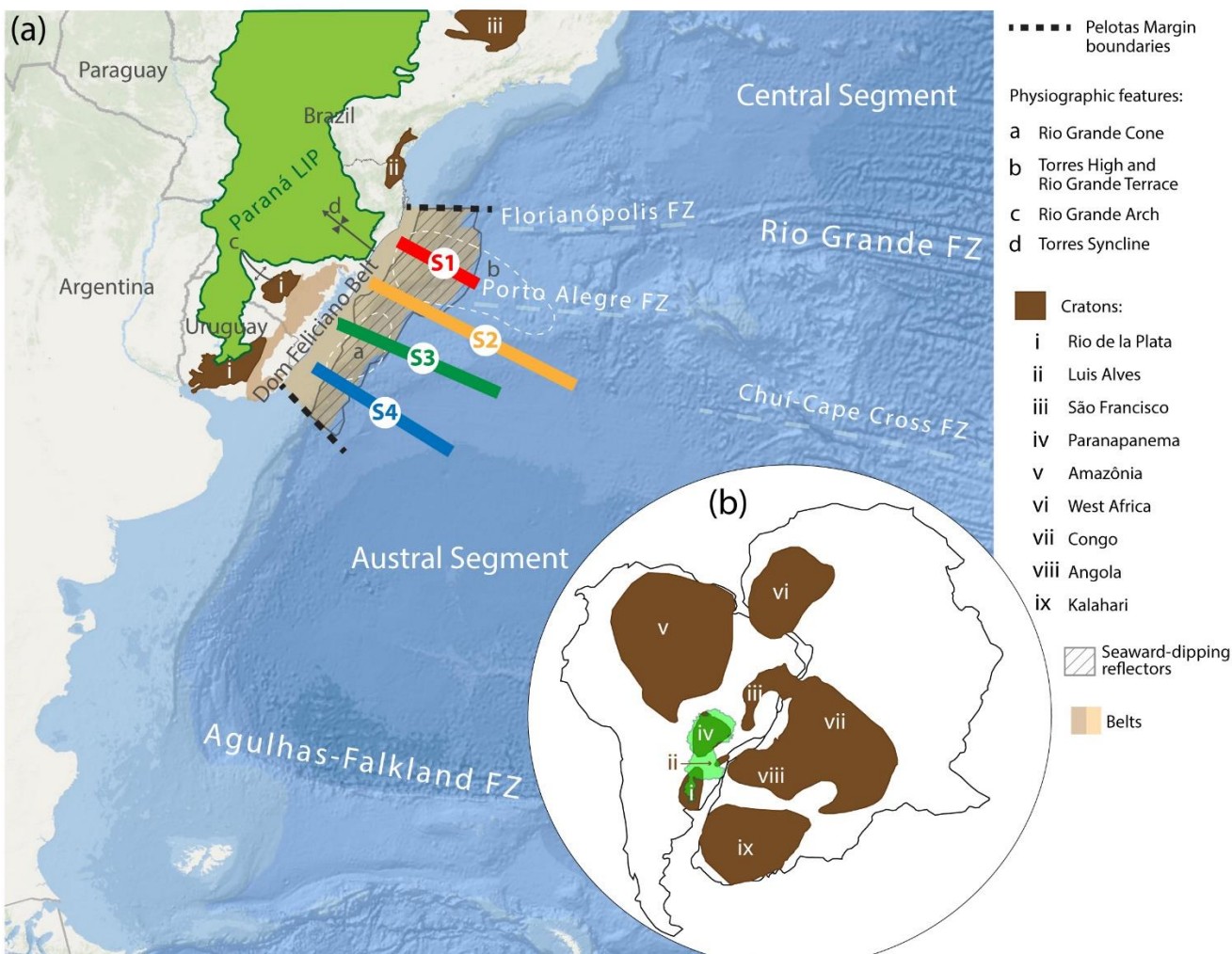

**Figure 1: (a) Map of the Austral South Atlantic (adapted from Cassel et al., 2022) showing: the location of the 4 Pelotas Margin seismic reflection profiles examined in this study; the distribution of seaward-dipping reflectors from Chauvet et al. (2021); crustal**
**basement from Stica et al. (2014); and Paraná Large Igneous Province (LIP) adapted from Rossetti et al. (2018). (b) Regional paleo-map of Western Gondwana adapted from Heilbron et al. (2008) showing the Paraná LIP and cratons.**

## 2 Geological setting

The Pelotas margin resulted from the assumed magma-rich breakup and separation of the Pangea super-continent during the Early Cretaceous. It is located offshore southern Brazil and is underlain by basement belonging to SW Gondwana. The continental basement is made of granitoids, schists and high-grade metamorphic rocks inherited from the Proterozoic Dom Feliciano Belt that records successive subduction and collision phases related to terrane accretion responsible for a strong NE-SW trending fabric (Chemale, 2000). The overlying pre-breakup sedimentary succession was deposited in the intracratonic Paleozoic Paraná Basin, which is capped by a continental, fissural magmatism of the Paraná Large Igneous Province (Serra Geral Formation) (Rossetti et al., 2018). These Lower Cretaceous flood basalts correspond to the Paraná-Etendeka Large Igneous Province that is tightly linked with the breakup of the Austral segment of the S-Atlantic (Zalan, 2004; Stica et al., 2014).

The Pelotas margin formed during the breakup of West Gondwana leading to the formation of the South Atlantic. This breakup may be regionally divided into Equatorial, Central and Austral segments with the Pelotas margin belonging to the latter (Stica et al., 2014). Stica et al. (2014) present a detailed compilation of the rift and breakup evolution of the Pelotas and conjugate Namibia margins. There is consensus that the Austral South Atlantic opening is diachronous, starting in the south and migrating northward (Franke et al., 2007). It is generally considered that final rifting and breakup of the Pelotas margin occurred in the Lower Cretaceous, with massive magmatic activity and the emplacement of high volumes of volcanic rocks forming prominent SDR sequences. The syn- to post-breakup sedimentary infill of the Pelotas margin can be subdivided into three main mega-sequences: i) transgressive mega-sequence (Aptian-Turonian), what includes the final rift phase, with depositional environments grading from continental deposits including alluvial conglomerates and lacustrine facies, to shallow marine evaporite, carbonate and siliciclastic facies deposited during breakup; ii) an aggradational mega-sequence (Turonian-Priabonian) with clastic fans in the more proximal domain and deeper marine shales and siltstone interbedded with turbiditic deposits in the distal domain; and iii) a regressive mega-sequence starting in Oligocene time and lasting to present, made of clastic fans and deltas that prograde oceanward over the distal deposits, forming a large regressive sedimentary wedge (Abreu and Anderson, 1998).

Recently, Cassel et al. (2022) demonstrated how along-strike variations in tectonic domains along the Andean convergence zone respond to the South Atlantic Mid Ocean Ridge spreading rate and control the margin evolution. While many of the previous studies have focused on the down dip and along strike magmatic evolution of the margin (Stica et al., 2014; Chauvet et al., 2021), little is known about the along-strike variations of the post-rift sediment accommodation and sediment-architecture and how it is linked to the volcanic addition. Here we focus on investigating the link between lateral variations of volcanic additions (e.g., SDR sequences) and the subsequent development of accommodation space and sedimentary infill along the Pelotas margin.

## 3 Along strike variation of volcanic addition and post-breakup sediment thickness

In this study, we interpret four parallel long-offset time-domain seismic reflection sections provided by TGS whose location
is shown in Fig. 1. We identify in the seismic sections the units: a) basement, b) SDR package, and c) sedimentary package.
These units are bounded from top to bottom by the seafloor, top of SDRs, base of SDR and Moho.

The basement unit is characterized by chaotic, discontinuous, low-amplitude reflectors (Fig. 2). Lines S1 and S3 (Fig. 2a and
Fig. 2b) continentward of distance 100 km show that top continental basement tapers down to about 9s TWT. Oceanward of
100 km, top basement remains parallel to the distal end of the seismic sections. The top basement interface is a smooth horizon
onto which the SDRs down-lap. The SDR package is characterized by several sequences of oceanward dipping, oceanward
diverging, high-amplitude reflectors. SDRs are well expressed on line S1, forming a thick volcanic package tapering
oceanwards and overlying both the tapering and the box shaped (i.e. uniform thickness) basement. In contrast, for line S3, the
SDRs are thin and overly only the crustal taper. The interface topping the SDR package corresponds to a sharp, high amplitude
reflection interpreted to separate magmatic extrusives from the post-rift sedimentary package. The latter is well stratified, their
reflectors have good lateral continuity and high frequency, showing parallel, sub-parallel, oblique and sigmoidal internal
pattern.

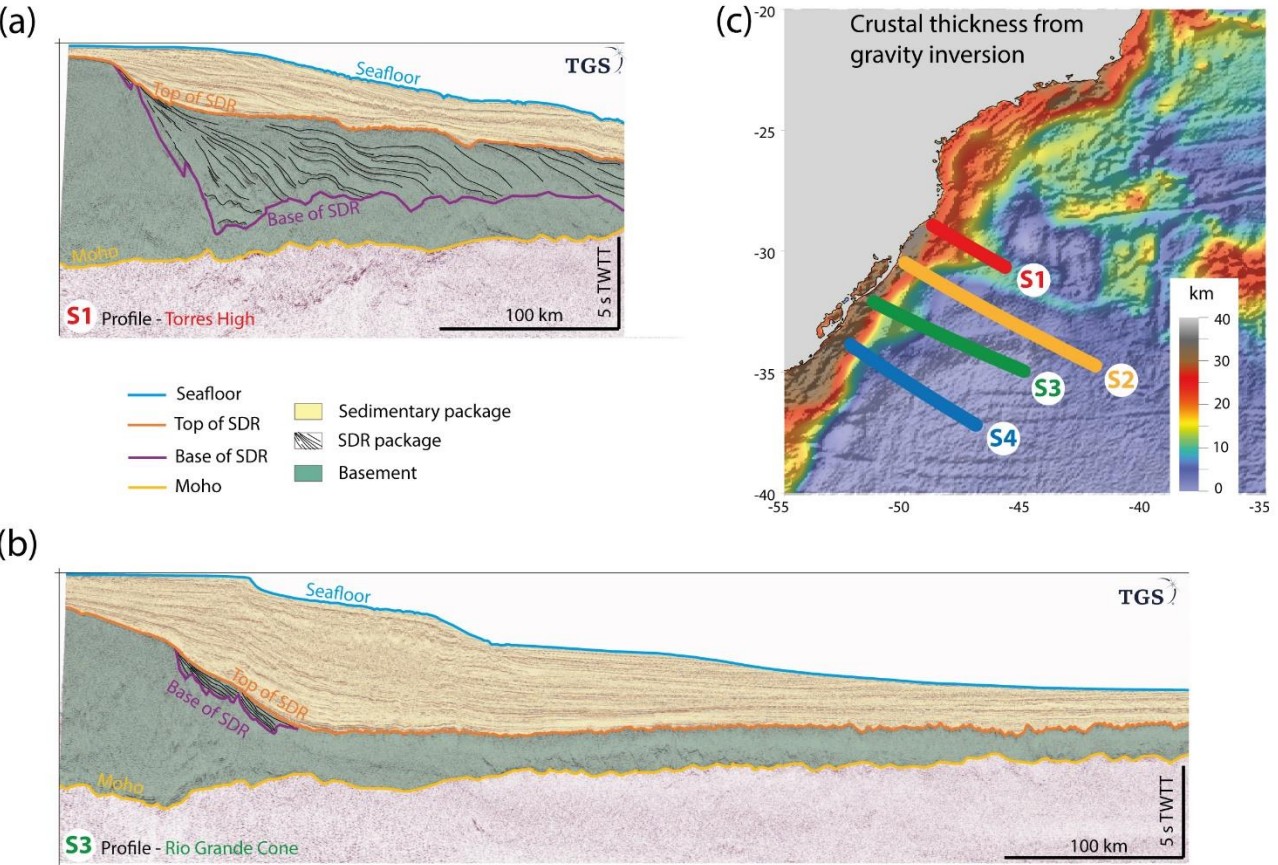

**Figure 2: (a) Seismic profiles in TWTT showing interpreted surfaces and units for profile S1 (b) Seismic profiles in TWTT showing interpreted surfaces and units for profile S3. (c) Locations of the 4 seismic profiles superimposed on map of crust thickness from gravity inversion (adapted from Graca et al. 2019): Profile S1 is located in the northern Pelotas margin along the Torres High. Profile S3 is located in the southern Pelotas margin crossing the Rio Grande Cone.**

Comparison of the four seismic sections in a regional along-strike perspective (Fig. 3) shows some major differences. While in line S1 the continent ward termination of the SDR sequence starts at approximately 30 km at about 2s TWTT, in other sections the SDR package starts further oceanward at about 4 s TWTT (Fig. 3). The oceanward termination of the SDR package occurs, except for line S1, at the inflection point of top basement, i.e., at the change from a tapering to a box shaped basement. Strong along-strike variations in the thickness of the volcanic (SDR) and sedimentary packages are shown in Table. 1 and Fig. 4.

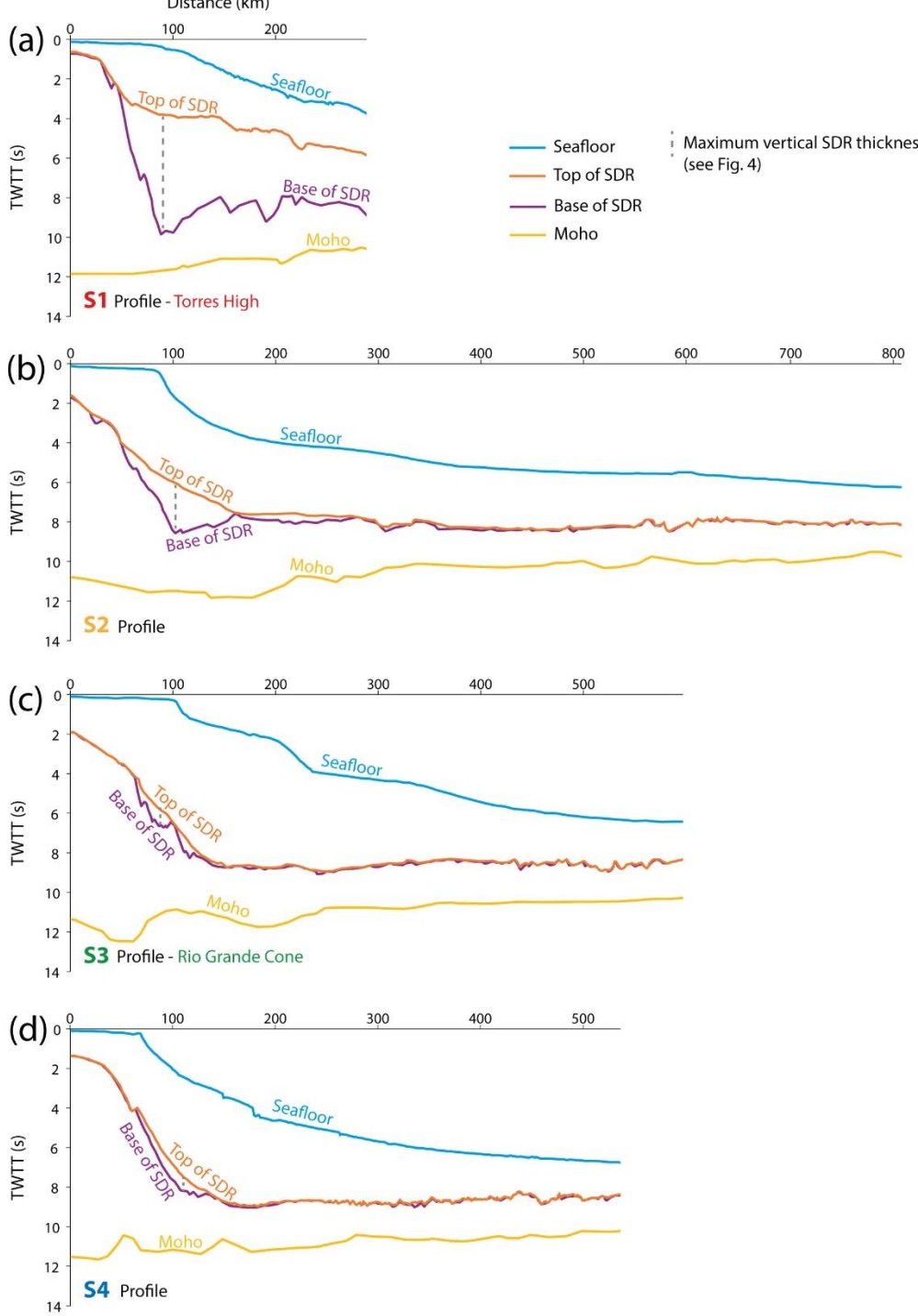

**Figure 3: Comparison of the 4 seismic profiles showing the interpreted surfaces in TWTT. (a) Seismic profile S1, located in the northern Pelotas margin crossing the Torres High. (b) Seismic profile S2. (c) Seismic profile S3 located in the southern Pelotas margin crossing the Rio Grande Cone. (d) Seismic profile S4.**

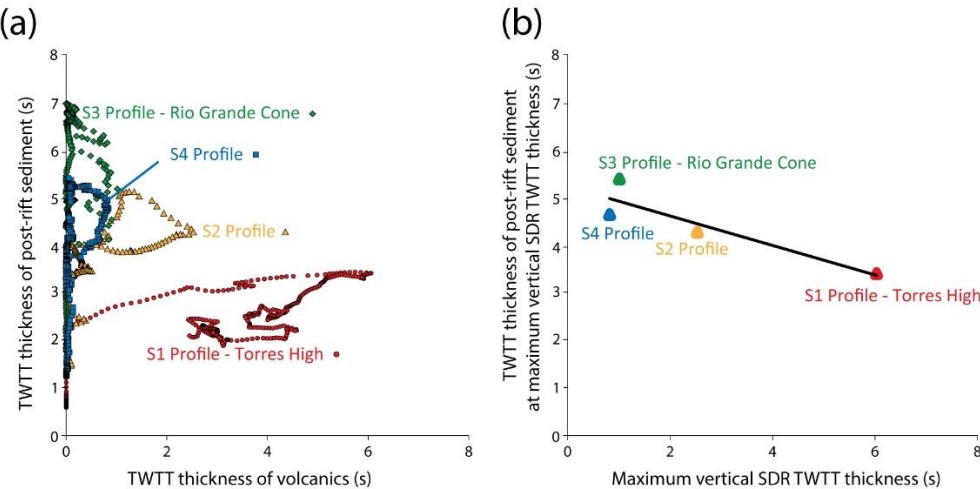

**Figure 4: (a) Plot of TWTT thickness of post-rift sediment against TWTT thickness of SDR for the same 300 km profile horizontal distance for each of the 4 profiles S1-S4. (b) Plot of TWTT thickness of post-rift sediment at maximum vertical SDR TWTT thickness against maximum vertical SDR TWTT thickness at the same horizontal profile distance for each profile S1-S4.**

**Table 1. Summary of vertical thickness measurements in TWTT taken from the Profiles S1 – S4, as shown in Fig. 3.**

| Profile | Maximum vertical thickness of SDR (s, TWTT) | Maximum vertical thickness of overlying sedimentary package (s, TWTT) | Ratio between vertical SDR and sediment thickness |
|---------|---------------------------------------------|------------------------------------------------------------------------|----------------------------------------------------|
| S1 | 6.06 | 3.80 | 1.59 |
| S2 | 2.53 | 4.00 | 0.62 |
| S3 | 1.01 | 6.00 | 0.16 |
| S4 | 0.84 | 5.80 | 0.14 |

Although all four seismic sections (S1-S4) show volcanic SDR packages, the thickness of volcanics and post-breakup sediments show notable changes in vertical thickness along strike. Figure 4a shows a plot of vertical sediment thickness TWTT against corresponding SDR thickness TWTT for each profile out to 300 km distance. It shows a clear difference between the value ranges of SDR and sediment thickness between the northern profile S1 (with high SDR and low sediment thickness) and the southern profiles S3 and S4 (with low SDR and high sediment thickness). Profile S2 shows an intermediate relationship. The relationship between maximum SDR TWTT thickness and the corresponding sediment TWTT thickness for each profile is shown in Fig. 4b. An inverse relationship can be seen; as volcanic (SDR) thickness in TWTT increases, the corresponding sediment TTWT decreases.

## 4 Variation of post-breakup accommodation space and dependency on volcanic addition

In the previous section, we observed an inverse correlation of post-breakup sediment thickness with volcanic addition. However, margin sediment thickness is dependent not only on accommodations space but also on sediment supply which is controlled by factors external to margin formation. In this section we determine the water-loaded post-rift accommodation space so that we can observe its relationship to volcanic addition. This requires flexural backstripping and decompaction to be applied to depth converted sections.

Figures 5a and 5c show the depth converted seismic interpretations for the Torres High and the Rio Grande Cone profiles shown in Fig. 2. The depth conversion for post-rift sediment thickness uses a depth-dependent seismic velocity function $V(z) = V_o + k.z$ where z is depth in km, $V_o = 1.75$ km/s and $k = 0.3$ km/s/km. Figure 8d in McDermott et al. (2019) shows k values between 0.4 and 0.5 km/s/km for sediments immediately above top SDRs. However, at depth, these k values produce an unrealistically high interval seismic velocity for profiles S2, S3 and S4 with thick sediment, hence we use a lower values of $k = 0.3$ km/s/km. Decreasing the k values results in a lower depth-converted thickness of post-rift sediment, which in turn results in lower estimate of post-rift accommodation space. Our calculation of post-breakup accommodation space is therefore a conservative lower estimate.  For simplicity we used 6.5 km/s interval seismic velocity for depth converting SDRs for all profiles (S1, S2, S3, S4) to generate the depth sections shown in Fig. 5. McDermott et al. (2019) show a laterally variable "skin" of lower interval seismic velocity SDRs about 2 km thick above deeper SDRs with 6.5 km/s. The average interval velocity for the whole SDR pile is therefore slightly less than 6.5 km/s. Because we only backstrip the post-breakup sediments (and not the SDRs), SDR thickness has no influence on the determined water-loaded post-breakup accommodation space.

Errors in the depth conversion of post-rift sediment thickness does affect the magnitude of water-loaded accommodation space determined by flexural backstripping and decompaction. However these errors are likely to be consistent between profiles so that the relative differences in the determined accommodation space between profiles, and our overall observation and interpretation are not changed. The uncertainty of seismic velocity required for depth-conversion highlights why we focus in figures 4 and 8 on measurements in TWTT; the primary seismic reflection observation is in TWTT while a depth-conversion is a model with often substantial uncertainty.

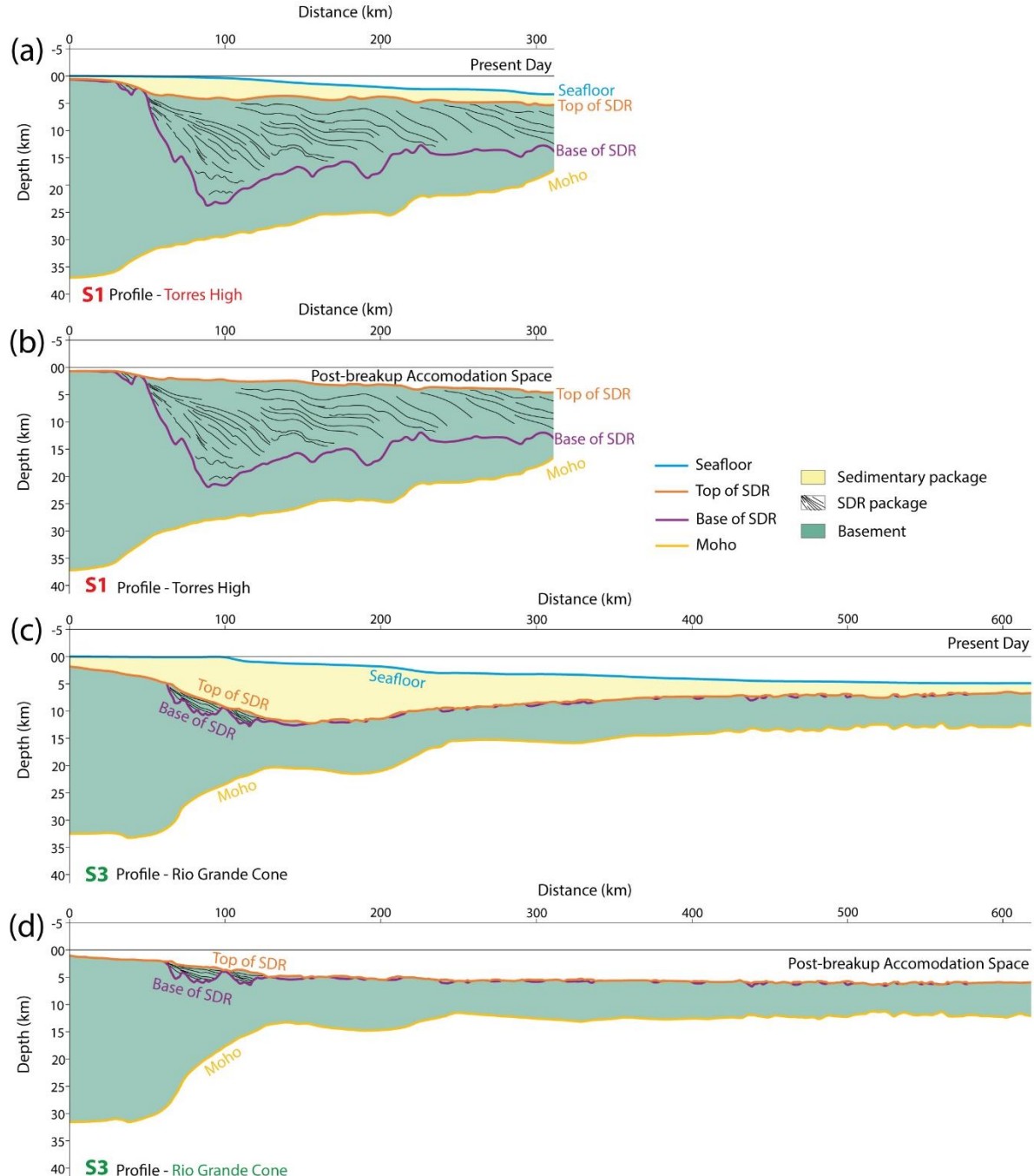

**Figure 5: (a and c)** Comparison of present-day depth-converted sections S1 and S3 showing the interpreted surfaces and corresponding units. **(b and d)** Comparison of water-loaded post-breakup accommodation space from flexural backstripping for profiles S1 and S3.

Post-rift accommodation space has been determined from the depth converted sections using 2D flexural back-stripping. This process consists of calculating the isostatic load of sediments and the consequent isostatic lithosphere rebound resulting from removal of that load. This isostatic rebound is applied to the top basement depth to determine the bathymetry that would exist at present if no post-rift sedimentation had occurred. Note that the result of flexural backstripping and decompaction is not a restoration to base post-rift; reverse post-rift thermal subsidence is not included. A detailed description of the 2D flexural

backstripping methodology is given in Kusznir et al. (1995) and Roberts et al. (1998). The magnitude of the sediment load depends on the thickness of sediment and density increase with depth due to compaction. We assume that the post-rift sediments are normally pressured and have a shaly-sand lithology. Compaction parameters for a shaly-sand have been used (Sclater and Christie, 1980). The SDRs are assumed to have experienced negligible compaction. A Te = 3km has been used to define the flexural strength of the lithosphere for the flexural back-stripping for removal of post-rift sediment loading

(Roberts et al., 1998). Sensitivity tests to Te are shown in Fig. S1.

The resulting water load accommodation space for the Torres High and Rio Grande Cone profiles are shown in Fig. 5b and Fig. 5d. These are directly compared in Fig. 6. For the same lateral position, the Rio Grande Cone profile shows significantly more accommodation space than the Torres High profile. While the predicted accommodation space is sensitive to the Te value used in the flexural backstripping, the significant difference between accommodation post-rift space for S1 and S3 profiles

remains.

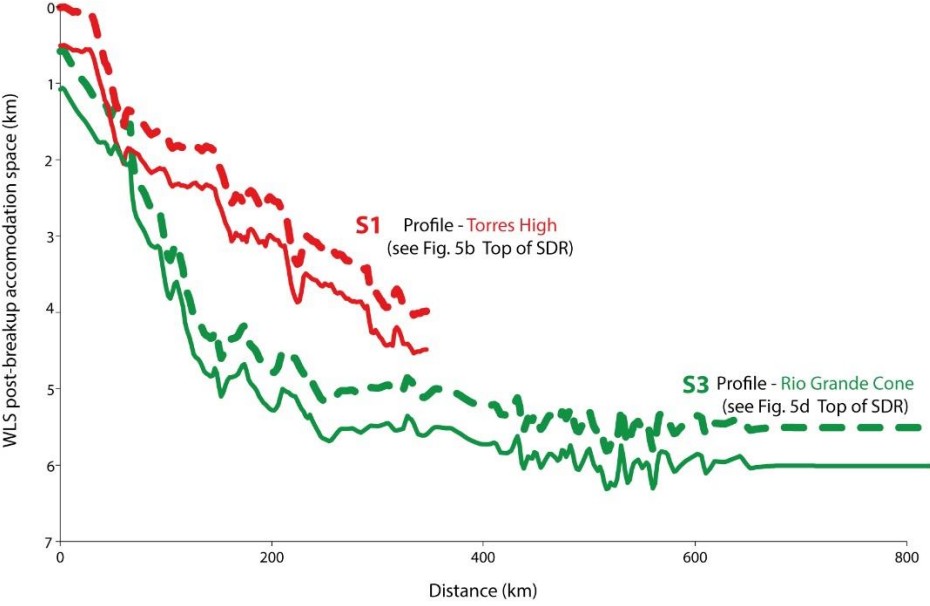

**Figure 6: Comparison of the water-loaded accommodation space from flexural backstripping (solid line) for the profiles S1 (Torres High) and S3 (Rio Grande Cone). Corresponding water-loaded accommodation space corrected for Oligo-Miocene dynamic subsidence (dashed line).**

The southern South American continental margins and adjacent ocean basins, including the Pelotas margin segment, experienced significant subduction dynamic subsidence in the Cenozoic as a consequence of Andean subduction of Nazca

oceanic lithosphere (Martinod et al., 2010; Shephard et al., 2012). This recent dynamic subsidence also contributes to the present-day water-loaded post-rift accommodation space. A correction of 500m for present day dynamic subsidence (a probable under-estimate) decreases the component of post-rift accommodation space attributable to Cretaceous continental
breakup. This component, directly related to the formation of the margin, when corrected for subduction dynamic subduction, is almost twice as large for the magma-normal southern profile (Rio Grande Cone) than the magma-rich northern profile (Torres High).

The along strike variation in post-rift accommodation space corrected for Oligo-Miocene subduction dynamic subsidence is shown in Fig. 7. Both Fig. 6 and Fig. 7 show that post-rift accommodation space increases substantially from north to south.
This anti-correlation with the decrease in volcanic addition observed from north to south is shown in Fig. 3 and Fig. 4.

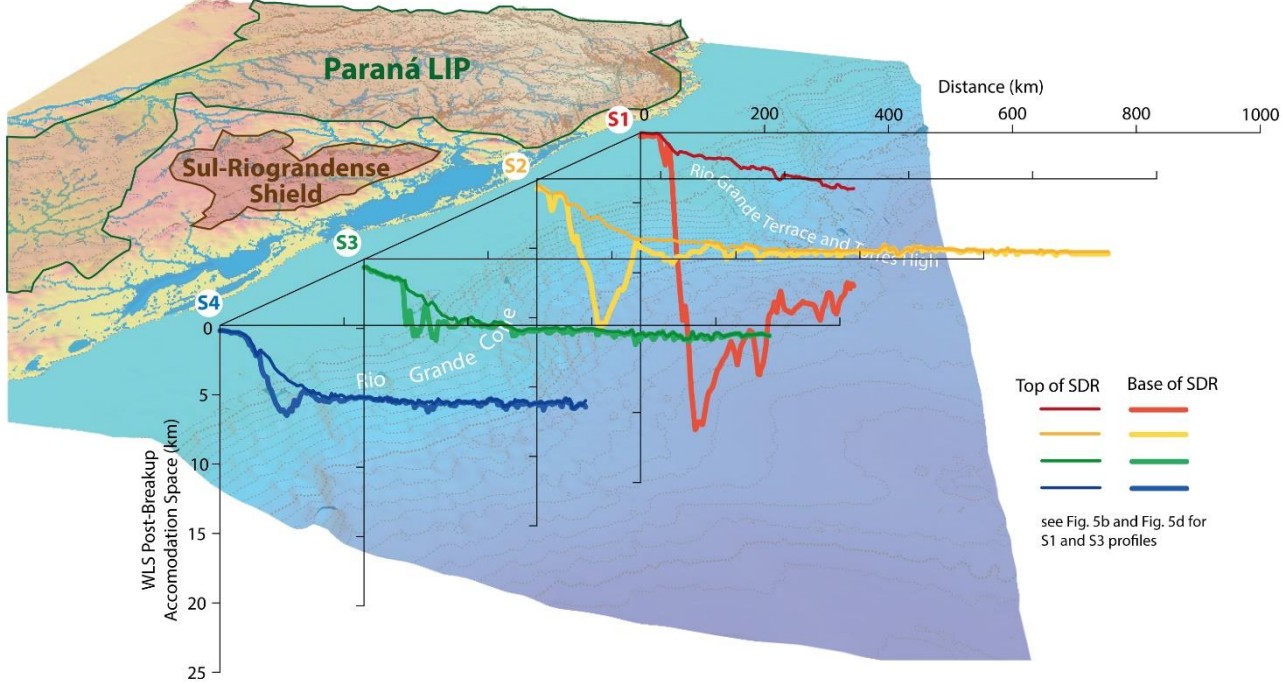

**Figure 7: Comparison of water-loaded post-rift accommodation space for the 4 profiles S1-S4 showing north to south variation along the Pelotas margin. Profiles S1 (Torres High) is offshore Serra Geral volcanics of the Parana LIP. Profiles S3 is offshore cratonic lithosphere of the Sul Riograndense Shield (SRS) where Serra Geral is absent.**

**5 Discussion**

**5.1 Along strike variation of magmatic addition along the Pelotas margin**

The Austral segment of the South Atlantic margin of South America is often assumed to be magma-rich along its whole length, however our analysis of the seismic reflection sections 3 and 4 (Fig. 3) demonstrate that this is clearly not correct.  While the northern profile S1 along the Torres High shows very large thicknesses of volcanic addition indicated by SDRs package up to

20 km thick, the southern profiles S3 and S4 across the Central and South Pelotas margin segments display magmatic thicknesses more consistent with those of a normal margin with oceanic crustal thickness ~ 6.5 km (Bown and White, 1994; Dick et al., 2003).

Total magmatic addition on a rifted margin consists of the sum of magmatic intrusives emplaced within and at the base of thinned continental crust (often termed magmatic underplate) and volcanic extrusives. It is not possible to reliably quantify

magmatic intrusives using seismic reflection and refraction data because their geophysical properties are similar to those of lower continental basement rocks (Karner et al., 2021). In our analysis we use the thicknesses of volcanic extrusives (SDRs) as a proxy for total magmatic volume. Estimates of the ratio of volcanic extrusives to magmatic intrusives/underplate range from approximately 1:2 for the Faeroes and Hatton Bank volcanic margins (White et al., 2008) to 2:1 for the Demerara Plateau (Gomez-Romeu et al., 2022). In all cases, measured thicknesses of volcanic extrusives represent a lower bound of total

magmatic volumes.

The north to south variation along strike of volcanic addition seen in Fig. 3 (in TWTT) and Fig. 7 (in depth) can be summarized by plotting maximum volcanic (SDR) interval TWTT against latitude. This north to south variation is shown in Fig. 8a and illustrates that the Pelotas margin is clearly not uniformly magma-rich. Within a distance of less than 300 km, volcanic addition varies from extremely magma-rich with SDRs up to 20km thick on the Torres High profile S1 to magma-normal for the Rio

Grande Cone profile S3 in the south.

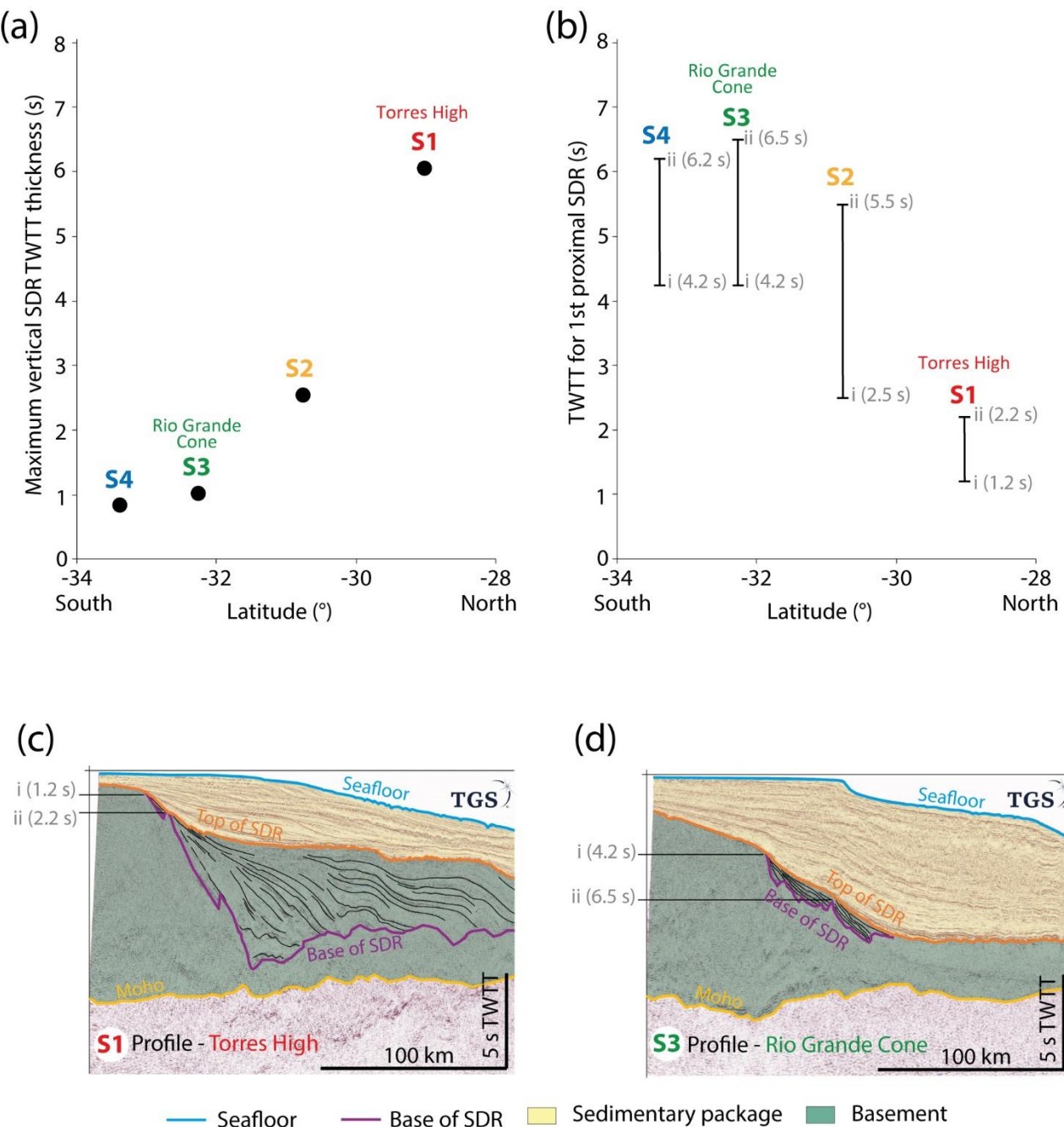

**Figure 8: (a)** Maximum thickness of SDRs in TWTT for profiles S1-S4 plotted against latitude showing north to south decrease. **(b)** TWTT of first proximal occurrence of SDRs plotted against latitude showing north to south decrease. **(c and d)** Comparison of the seismic reflection sections for S1 and S3 in TWTT highlighting the first proximal occurrence of SDRs. Two measurements of the TWTT of first proximal SDRs are shown in Figures 8b, c and d; (i) denotes the lower value, (ii) denotes the upper value. See text for more detailed explanation.

This large variation in extrusive magmatic volumes along strike along the Pelotas margin is consistent with the observation reported in Sauter et al. (2023). This variation also correlates with the distribution of Serra Geral volcanics (Paraná LIP) on land (Fig. 1 Fig. and 7). Profiles 3 and 4, with normal magmatic volumes, are located offshore to where the Serra Geral is absent. In contrast, profile S1, which shows the very large SDR thicknesses on the Torres High is located offshore where the Serra Geral is very thick and reaches the coast. The absence of Serra Geral in central and southern Rio Grande do Sul coincides with the presence of cratonic lithosphere of Sul Riograndense Shield (Chemale, 2000). In contrast, on land to the north, the distribution of Serra Geral coincides with that of the Palaeozoic Parana Basin. As discussed in Sauter et al. (2023), this observed rapid decrease in magmatic volumes along strike may suggest that the large magmatic volumes observed in the north of the Pelotas margin (e.g., Torres High) are generated by a component of mantle inheritance rather than the usually assumed hot mantle-plume mechanism alone.

**5.2 Along strike variation of accommodation space as consequence of magmatic addition**

Examination of seismic reflection profiles S1-S4 indicates that there is an inverse correlation of sediment TWTT with SDR TWTT thickness (Fig. 3) as shown in the cross-plot in Fig. 4. Sediment thickness is controlled by many factors including source area erosion, sediment transport, deposition and preservation. As a consequence, we prefer to examine the lateral along strike variation in accommodation space rather than sediment thickness. Post-rift (post-SDR) accommodation space, determined using 2D flexural backstripping, shows large variations along strike (Fig. 5, Fig. 6 and Fig. 7), which inversely correlates with the thickness of extrusive volcanics.

During rifting leading to continental breakup, syn-rift subsidence occurs in response to thinning of the continental crust, which is partly offset by thermal uplift from geotherm elevation (McKenzie, 1978). After breakup, re-equilibration of the elevated geotherm results in post-rift thermal subsidence. The amount of accommodation space available for post-rift sedimentation depends on the sum of accommodation space generated by post-rift thermal subsidence and that remaining unfilled from the syn-rift stage.

In the north of the Pelotas margin, where magmatic addition was very large, syn-rift accommodation space was filled by extrusive volcanics producing almost 20 km of SDRs on the Torres High (profile S1). These SDRs have long lateral flow lengths, which are interpreted to indicate that the top of the SDRs were deposited at or above sea-level (Mutter et al., 1982; Planke et al., 2000). As a consequence, the accommodation space available for post-rift sedimentation observed today and shown in Fig. 5, Fig. 6 and Fig. 7 consists of only that generated by post-rift thermal subsidence. In contrast in the south of the Pelotas margin (profiles S3 and S4), where magmatic addition is much less, syn-rift accommodation space was underfilled providing an additional contribution to add to accommodation space generated by post-rift thermal subsidence. As a consequence, more accommodation space is available in the south of the Pelotas margin for sediment deposition above volcanic extrusives. The observed inverse correlation of accommodation space with the thickness of extrusive volcanics can therefore be explained by the control of residual syn-rift accommodation space by the volume of extrusive volcanics. Put simply, syn-rift accommodation space filled by extrusive volcanics is no longer available for post-rift sedimentation.

The Pelotas margin has two major present-day offshore physiographic features; the Torres High in the north (imaged on profile S1) and the Rio Grande Cone in the south (imaged by profile S3). The former exists because of magma-rich breakup generating very thick SDRs, the latter is located where the breakup occurred with much less magmatic addition provided a larger amount of accommodation for thick post-rift sedimentation. Both physiographic features control oceanic drifts by deflecting ocean currents but themselves have different origins. The variation in magmatic addition along the Pelotas margin exerts a strong control on depositional environments.

### 5.3 Significance of TWTT depth of first proximal volcanics on seismic reflection sections

Examination of the seismic reflection sections in the time domain shows that the TWTT for the first appearance of proximal SDRs is also very variable. The observed north to south variation of TWTT of first volcanics is plotted as a function of latitude in Fig. 8b. For the purpose of showing the uncertainty in the measurement, two measurements of the TWTT of first proximal SDRs are plotted for each profile corresponding to lower and higher measured values. The term SDR *sensu stricto* simply means "seaward dipping reflector", and while the common use of the term is applied to volcanic seaward dipping reflectors, seaward dipping reflectors can also be generated by sedimentary sequences within fault controlled half-grabens. The lower TWTT values shown in Fig. 8b may correspond to either the onset of volcanics or sedimentary accumulations; their exact nature cannot be reliably determined using the available seismic data alone. In contrast the higher TWTT values shown in Fig. 8b represent a much more certain measurement for the onset of volcanics. The TWTT of first volcanic SDRs, using either lower or higher measured values, shows an inverse correlation with the magnitude of volcanic addition shown in Fig. 8a.

The lower and higher measurements of first proximal SDRs are shown in Fig. 8c for the magma-rich margin profile S1 over the Torres High in the north, and in Fig, 8d for the magma-normal margin profile S3 in the south. For profile S1, TWTT measurements lie in the range 1.2 to 2.2 s. These values are similar to those reported by Mutter et al. (1982) and Planke et al. (2000) for the long flow-length SDRs on the Voring segment of the Norwegian margin which formed at or above sea-level and have subsequently thermally subsided. For profile S3, TWTT measurements lie in the range 4.2 to 6.5 s similar to those reported by Planke et al. (2000) and Hinz et al. (1999) for the deep marine erupted SDRs on the Exmouth Plateau the Argentine margins.

We explore this observed inverse correlation using a simple isostatically balanced model of a rifted margin with varying amounts of magmatic addition. The simple model, described in more detail in Chenin et al. (2023), calculates the isostatically balanced crustal cross-section for the idealised rifted margin produced by a prescribed continental crust and lithosphere thinning taper showing resulting bathymetry, the remaining thickness of the continental crust and the thickness of new magmatic addition. The amount of decompression melt is calculated from the thinning factor taper using a parameterisation of the decompression melt model of White and McKenzie (1989). Isostatically balanced cross-sections are produced for thermally re-equilibrated lithosphere and at syn-breakup time by including lithosphere thermal uplift from the syn-tectonic elevated geotherm consistent with McKenzie (1978). Magmatic addition is partitioned 1/3 as extrusives overlying the thinned continental crust and 2/3 intrusives (underplate); in the oceanic domain these two layers correspond approximately to oceanic

layers 2 and 3. The model is used to examine the magma-rich or magma-poor consequences for margin architecture and accommodation space resulting from increasing or decreasing the amount of decompression melt with respect to the 7 km generating normal oceanic crust and also the timing of melt initiation with respect to crustal thinning.

Figure 9 shows isostatically balanced margin cross-sections at breakup (with thermal uplift) and full thermal re-equilibration for an idealized margin with normal magmatic addition (left) and magma-rich addition (right). The magma-normal model assumes a maximum of 7 km magmatic addition (forming normal thickness oceanic crust) with decompression melting starting at $\beta = 3$ consistent with the decompression melt model of White and McKenzie (1989). The magma-rich model has a maximum magmatic addition of 10 km (producing a 10km oceanic crust) with decompression melting starting at $\beta = 2$ with the onset of decompression melting slightly advanced with respect to crustal thinning.

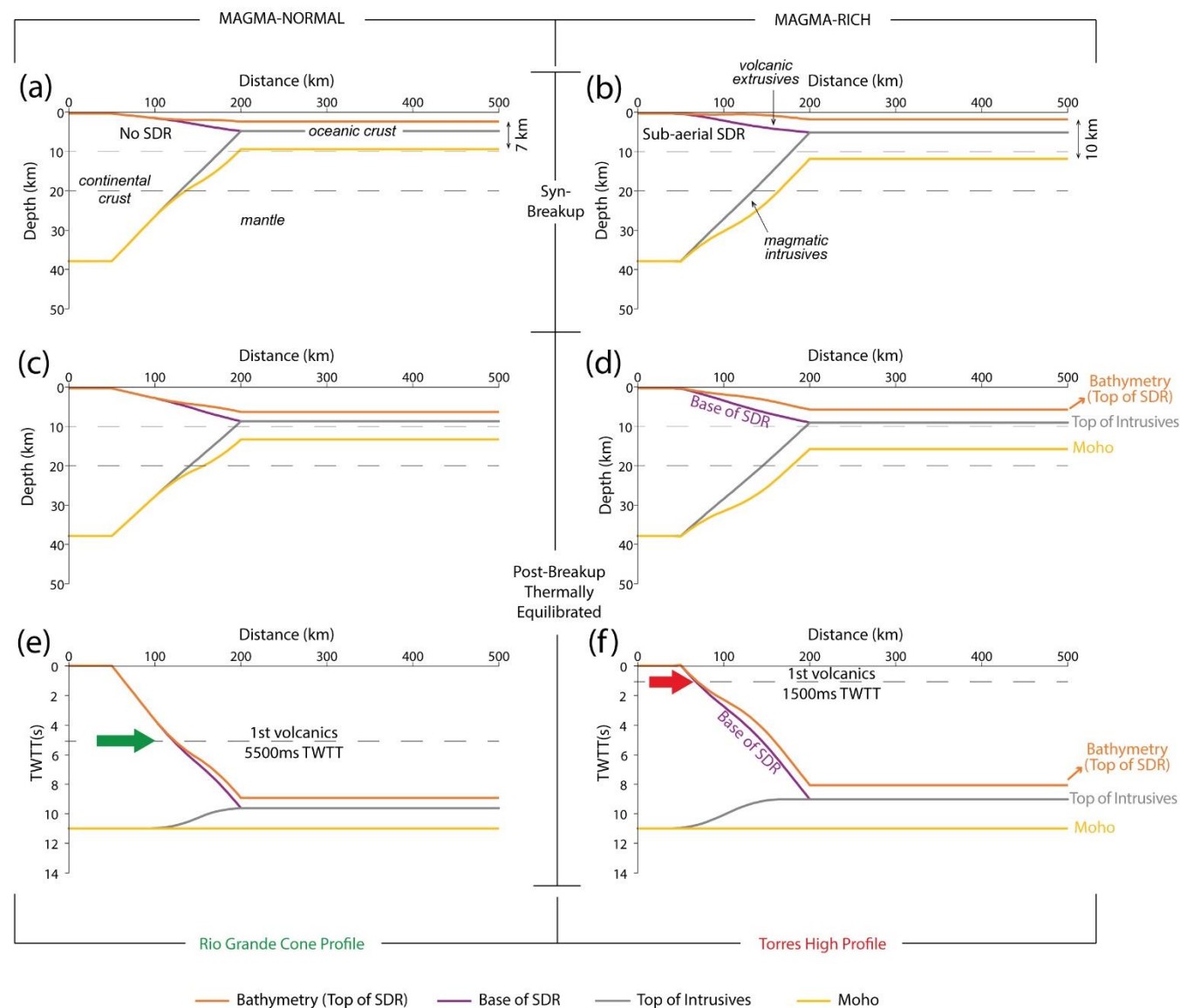

**Figure 9: Isostatically balanced model cross-sections of a rifted margin with varying amounts of magmatic addition. (a and b) at breakup time with syn-rift lithosphere thermal uplift. (c and d) at large post-breakup time with full lithosphere thermal re-equilibration. (e and f) Time domain representation of model cross-sections for full lithosphere thermal re-equilibration converted from depth model using Warner's "10 second rule" (1987). The proximal onset of first volcanics is indicated by the coloured arrows. Left column: an idealized margin with normal magmatic addition as for the Rio Grande Cone S3 Profile. Right column: an idealized magma-rich margin as for the Torres High S1 Profile.**

At breakup the magma-rich model shows (Fig. 9b) the upper surface of proximal volcanics at or above sea-level consistent with SDRs with long flow lengths as seen on the Torres High seismic section S1. In contrast the normal magmatic addition model shows (Fig. 9a) the upper surface of first volcanics at ~ 2 km water depth corresponding to submarine lava flows erupted onto thinned continental crust and consistent with the observation that the top of distal deep-water SDRs often merges

seamlessly with the top of oceanic layer 2 (Hinz et al. 1999). The water depth of first volcanics is controlled by the isostatic consequences of the relative timing of crustal/lithospheric thinning and the onset of melt production by decompression melting

(see Chenin et al. 2023). Early melt production relative to crustal/lithospheric thinning reduces the bathymetry of first volcanics. Factors advancing the initiation of melt production with respect to crustal/lithospheric thinning are elevated lithosphere and asthenosphere temperature, inherited lithosphere chemical enrichment or lithosphere deformation mode (Lu and Huismans, 2021). The corresponding cross-sections after thermal re-equilibration and subsidence are shown in Fig. 9c and Fig. 9d.

Warner (1987) observed that the Moho TWTT on marine deep long-offset seismic data was consistently at about 10 s TWTT for thermally equilibrated lithosphere and was remarkably constant (flat) in time irrespective of the complexity of the geology above including sediment thickness variation. Invoking Warner's 10 s rule for the Moho TWTT for thermally equilibrated lithosphere allows the cross-sections shown in Fig. 9c and Fig. 9d to be converted into the time domain as shown in Fig. 9e and Fig. 9f. To do this basement thickness is converted to interval TWTT (using a basement velocity of 6.5 km/s) and

subtracted from 10s to give the TWTT of top basement; this estimate of TWTT of top basement is therefore independent of the interval TWTT of bathymetry and post-rift sediments. In the time domain, first volcanics are predicted to occur at ~ 1.5s TWTT for the magma-rich model, while for the magma-normal magmatic addition model first volcanics occur at ~ 5.5 s. These model predictions are consistent with the observed TWTT of first proximal volcanics between 1.2 to 2.2 s for the Torres High Profile S1 and 4.2 to 6.5 s for the Rio Grande Cone Profile S3 shown in Fig. 8. It should be noted that the model prediction

assumes a fully equilibrated lithosphere thermal structure while the Pelotas margin with Early Cretaceous breakup age is not yet fully re-equilibrated.

A common classification of rifted margins is whether a margin is magma-rich, magma-normal or magma-poor. An obvious approach to distinguishing a magma-rich from a magma-normal margin, or a margin's position in between these two end members, might appear to be through measurement of the volume of magmatic addition. However the problem arises, as highlighted by Tugend et al. (2020), that in practice thinned continental crust at a rifted margin cannot reliably be distinguished

from volcanic extrusives above it and magmatic intrusives (underplate) below for reasons explained by Karner et al. (2021). As a consequence it is not possible to accurately measure the volume of magmatic addition at a rifted margin. Tugend et al. (2019) and Chenin et al. (2024) also show that the relative timing of decompression melting with respect to crustal thinning may be as important as magmatic volume in generating a magma-rich margin. As shown above, the TWTT of first proximal

volcanics may represent a practical and efficient method of distinguishing a magma-rich form a magma-normal margin, or for placing a margin in between these two end-members.

**6 Summary**

• The amount of magmatic addition on the Pelotas margin varies substantially along strike from extremely magma-rich to magma-normal within a distance of ~300 km.

•	In the north, where the SDR package is thickest, the Torres High shows SDR thicknesses of ~ 20 km and post-breakup water-loaded accommodation space is much less than in the south where magmatic addition is normal and SDR thicknesses are small.

•	Post-breakup accommodation space correlates inversely with SDR thickness, being less for magma-rich margins and more for magma normal/intermediate margins.

•	The Rio Grande Cone is underlain by small SDR thicknesses allowing large post-breakup accommodation space and the accumulation of large sediment thickness.

•	The observed inverse relationship between post-breakup accommodation space and SDR thickness is predicted by a simple isostatic model of continental lithosphere thinning and decompression melting during breakup.

•	In the time domain, a magma-rich margin, with sub-aerial SDR flows, shows first volcanics   between 1.2 to 2.2 s
TWTT while a "normal" magmatic margin has first volcanics  between 4.2 and 6.5 TWTT.

•	Our study shows that the TWTT of first volcanics may provide an alternative approach for distinguishing magma-rich margins from margins with normal magmatic addition compared to estimating total magmatic volumes.

•	The methodology that we use in this paper provides a new approach for investigating the complex magmatic and sedimentary evolution of rifted continental margins.

**Author contribution**

MC: Conceptualization, seismic interpretation, data analysis, visualization, writing (original draft and revisions).  NK: Conceptualization, seismic interpretation, data analysis, methodology, modelling, writing (original draft and revisions) GM: Conceptualization, seismic interpretation, writing (review and revisions), funding.  DS: Conceptualization, data access, seismic interpretation, writing (review and revisions).

**Competing interests**

The authors declare that they have no conflict interest.

**Acknowledgements**

We thank TGS for the seismic data supporting this study. The raw data are the private properties of TGS who should be contacted for any lending or acquisition (https:// www.tgs.com/). We also thank colleagues to comments and discussion.

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
