# Peer review of "Along-strike variation of volcanic addition controlling post breakup sedimentary infill: Pelotas margin, Austral South Atlantic"

_EGUsphere, 2023_

## Referee Comment (RC1)

**General Comments**

In this manuscript, Cassel et al. present seismic data from the South Atlantic and discuss the along-strike variability of SDR and its correlation and influence on accommodation space. It is then shown that these results are consistent with variation in decompression melting during breakup.

Overall, the manuscript is novel, of clear scientific significance, useful for the community and well argued. The aim, the focus of the paper and the conclusions are clear. It is overall well written, and the figures are clear (and look nice!).

Nevertheless, there is still room for some improvement to improve the readability and strengthen the argumentation. Precisions or corrections are definitely needed for the 4s vs 6s TWTT of first volcanics and for the generation of the results of the model in time domain (see details below). Some interpretations might be restructured. Some precisions seem to be needed for some methods. There are several potential small improvements to the figures and the text. Finally, I also propose a series of comments that are more proposition than corrections and that the authors might elect not to follow.

After these improvements/precisions/corrections, I'm sure that the manuscript will be a great contribution to the understanding of passive margins and a great addition to Solid Earth.

**Specific Comments**

**Lines 134-135:** I'd say that these velocity values (for sediments) should be justified. A simple reference or a small sentence might be sufficient but at the moment they come out of nowhere. Or are they also coming from McDermott et al. 2019 like for the SDR?

**Lines 136-137:** "The SDRs of the Torres High profile are very thick and are most likely composed of basaltic flows. As a consequence, we use a higher velocity of 6.5 km/s for depth conversion." Did you use 6.5 km/s for all SDR of the study or only for the Torres High profile? From the phrasing, it is not clear to me. If other velocities where used, this should be stated. If the same velocity was used, you might also add a sentence to justify it as your observation "The SDRs of the Torres High profile are very thick" is only valid for the S1 profile. I agree that these changes have no influence on the conclusions of this paper but it would be scientifically nice to justify your choices (simplicity and comparability might be valid reasons on this one :-) ).

**Lines 234 ff:** I find that this part on the Rio Grande Cone is not the strongest of your manuscript. I'd say that the Rio Grande Cone is mainly there because a large river brought sediments there. And then the high accommodation space caused by the not-so-magmatic margin allowed it to deposit on top of the passive margin. But let's make a thought experiment. What if this river brought sediments over S1 (or S2) profile? The same delta would probably also have deposited, but just further offshore, on top of oceanic crust or the very distal passive margin. So I'd say that the main control on the presence of this delta is mainly the presence of the river and the high accommodation space allowed it to deposit not too far offshore. The structure of the margin (magma very rich or not so rich) does control the accommodation space on the margin but not further oceanwards where there is anyway plenty of space (except exactly on the Walwis ridge but that's not a simple SDR). Maybe you could rephrase it as "the high accommodation space allowed the Rio Grande cone to deposit large thicknesses of sediments on top of the margin". Anyway, I feel that this discussion on the Rio Grande Cone is not super strong or the most interesting of your paper and you might want to remove it from the abstract and/or summary (but not from the main text). What do you think?

**Line 243:** "for the magma-normal margin profiles in the south, first proximal SDRs occur at 6s or deeper". That is for me the main problem of the manuscript. Your figure 3 and 8 show that it occurs around 4s and not 6s. This is correctly mentioned in the Abstract "first volcanics are observed at 4.2s TWTT or deeper" but not here. This also has implications for the consistency of your model (section 5.3). And for your conclusion ("In the time domain, a magma-rich margin, with sub-aerial SDR flows, shows first volcanics at ~2s TWTT while a "normal" magmatic margin has first volcanics at 6 -7s TWTT."). It's not a huge deal as it doesn't change the conclusions of your paper, it's still deeper than 2s. But it's not 6s, that's incorrect.

**Section 5.3 (lines 241-281):**

I'm not 100% comfortable with this section. Maybe some things have to be restructured/rewritten or better explained. I see 4 problems/improvement potentials with this section (and maybe with the structure of the whole discussion). Here they are and I'll go more in details on each after.

1.  Methodology: I find strange to find methodological description here, I don't understand one part of the method and I'd argue that you need to include sediments to compare it with a real margin.
2.  The model shows 6-7s TWTT while the seismic 4s.
3.  I don't understand why the results of the modelling come only to support the TWTT of first volcanics and not to support also sections 5.1 and 5.2
4.  I don't understand why you focus only on TWTT of first volcanics to identify magma-rich/magma-normal margins.

Ok, let's go in the details of each comment.

**Comment 1:**

Lines 246 – 263 and 275-278: I have the feeling that this is more methods and should not be placed in the discussion chapter (5). I would make a new section before the discussion to present the methods. As a lazy reader, I often want to read the discussion chapter without the details of the method and I thrust the (lazy) reviewers to have checked the methods :-) . At the moment, I think it "dilutes" a bit your discussion points. What do you think?

Line 275 ff.: Here I don't get how you calculated your sections in time. Ok, you assumed that the Moho is at 10s (BTW on the figure it's a bit deeper than 10s) but how did you calculate the other reflectors? Did you use the velocities mentioned in section 4 and you used this 10s-rule just to compensate for the lack of good constrains on the velocity of the crustal basement or...? This needs to be better explained.

Also, how can you model a section in time and compare it with your seismic if you don't model post-rift sediments? I understand that the post-breakup thermally equilibrated sections (fig. 9 c and d) are just a concept and not used to compare with reality 1:1 and thus it's not a big problem not to model sediments. But you compare the time-converted sections (fig.9e, f) 1:1 with real-world examples. And I'd argue that the presence of sediments instead of water would have a big influence as the seismic velocities are completely different. Without a change, I'd say that the comparison is invalid. But as I did not fully understand your methodology, maybe I missed something.

As they are four problems with this time-converted section (two highlighted here, one in comment 2 and one in comment 4), you might elect, in an extreme case if you can't correct it, to just remove this time comparison and only use the modelling to show the change in volume of volcanics and changes in accommodation space (see also comment 3 and 4). The paper would still be strong, relevant and useful without this. But of course, it's nice to have.

**Comment 2:**

The model shows first volcanic material at 6-7s TWTT while the seismic shows it at 4s (as already discussed above). Maybe you have to re-run your model with decompression melting starting a bit earlier to match with your observations at 4s. Or what if you include sediments in your time-migrated model (as discussed in comment 1)? As they have velocities higher than water, that might pull up the appearance of the first volcanics and better fit your observations.

**Comment 3:**

I don't understand why your modelling results only come at this point of the manuscript. In the main text, you only use it to justify your 4/6s TWTT but in the abstract and the summary, you also use it to justify the difference in SDR thickness and accommodation space ("*The observed inverse relationship between post-breakup accommodation space and SDR thickness is consistent with predictions by a simple isostatic model of continental lithosphere thinning and decompression melting during breakup*" and "*The observed inverse relationship between post-breakup accommodation space and SDR thickness is predicted by a simple isostatic model of continental lithosphere thinning and decompression melting during breakup*").

I think you have to discuss it a little bit more in the text. You need to justify it in the text before presenting them as "conclusions". At the moment you only have one sentence "lost" in section 5.3 about accommodation space (line 271-273) but nothing about volume of volcanics. Of course, I agree with you, your model show it, but you should mention it clearly in the text. Why not use them in the text to justify your point 5.1 and 5.2? Your model indeed does not only show the difference in TWTT of first volcanics, but also that the thickness of volcanic material and accommodation space changes with the different parameters. Why only use it for 5.3, right at the end?

Your order of argumentation at the moment also makes less sense as you already concluded in section 5.1 that the margin is more magma-rich in the N. Why would you then test the same hypothesis with a model in section 5.3 if you already came to this conclusion? I see two options to circumvent this problem: either you present your modelling first (as already proposed on comment 1 and see further comments) and then can also use these results to support your point 5.1 and 5.2. Or you say that you don't need to prove this with a model and that the thickness is the volcanic sequences is enough to prove we are more magmatic (and I would agree with this). If so, you wouldn't need the modelling at all and you can discuss the TWTT of first volcanics based on real-world data alone (but I'd say that your model is still a good addition to the discussion). Maybe you see another solution to circumvent the problem?

**Comment 4**

I see another problem in the summary linked to section 5.3: "Our study shows that SDRs are not synonymous of magma-rich margins; the TWTT of first volcanics may provide a better approach to distinguishing magma-rich margins from margins with normal magmatic addition". Again, this has not clearly been discussed before coming to the summary. Your whole study (observation + model) showed that the thickness of volcanic material [and accommodation space as a consequence] also distinguishes magma-rich and magma-normal margins (and not only TWTT of first volcanics). Why do you only focus on the TWTT of first volcanics to determine whether it's a magma-rich or magma-normal margin in the summary and abstract? Is it because it's easier to measure than thickness of volcanics and accommodation space? That would be a good reason but should be discussed in the text.

Also, the age of the margin might also play a role here. Do you think that this boundary would also be at 4s TWTT on a very recent margin with almost no post-rift sediments yet and less thermal subsidence? I'd say no (although I do agree that after a while, thermal subsidence is close to 0 and the accommodation space above the first volcanics is anyway filled (as on your example)). In the case of a very young margin, the thickness of magmatic material would be the best parameter to determine between magma-rich and magma-normal margins. This brings us back to the previous paragraph: why is the TWTT of first volcanics a better method that other methods? This limitation should probably be discussed in the text.

**Suggestion of improvement for the structure of the discussion:**

As seen from the comments above, the reasoning path of the discussion section could be changed to something like:

We see several differences across the strike of the margin: thickness of volcanics, accommodation space and TWTT of first volcanism (2 vs. 4/6s) → We relate this to a change of magmatic production → Our model allow to test this → Thickness of volcanics, accommodation space and TWTT of first volcanics of the model fit real-world observations (story of 6 vs. 4s and inclusion of sediments in the model apart) → These differences in thickness, accommodation space and TWTT of first volcanics can be explained by a different volume and appearance time (or beta factor) of decompression melting (along with the other arguments you already give in section 5.1) → Both thickness of volcanics, accommodation time and TWTT of first volcanics provides a way to determine magma-rich margin more reliably than the presence of SDR. → Because TWTT of first volcanics is easier to measure, it probably provides the simplest way to determine magma-rich margin (or another argument as to why this parameter is important) → then a last section of the discussion with current section 5.2 about accommodation space (including that it is also confirmed by the model).

That seems like a big change but basically you can just copy and paste most of your existing text. But that's only just a suggestion to use your modelling to support all your points and make your discussion clearer, more relevant and more impactful. You might elect to not follow it at all, that's not a problem.

**Summary (lines 282-298):** As discussed before, you might consider removing the Rio Grande Cone story as it not the most important outcome of the study. Anyway, I'd maybe put the point "The observed

inverse relationship between post-breakup accommodation…" right after the point "Post-breakup accommodation space correlates inversely with SDR thickness…" as those are linked.

Line 296, this 6-7s has to be clarified. 6-7s is only from the model at the moment, not from seismic data.

Line 297-298: In light of the discussion above, the sentence could be slightly changed to "may provide the simplest approach to distinguishing …"

**Abstract:**

Maybe linked with comment 3 above, I'd also maybe reshuffle the order of the sentences of the abstract to put your model upfront [but this is just a suggestion]:

"We show that post-breakup accommodation space correlates inversely with SDR thickness, being less for magma-rich margins and more for magma normal/intermediate margins. *The observed inverse relationship between post-breakup accommodation space and SDR thickness is consistent with predictions by a simple isostatic model of continental lithosphere thinning and decompression melting during breakup.* [The Rio Grande Cone, with large sediment thickness, is underlain by small SDR thicknesses allowing large post-breakup accommodation space.] A relationship is observed between the amount of volcanic material and the TWTT of first volcanics; first volcanics are observed at 1.25s TWTT for the highly magmatic Torres High profile while, in contrast, for the normally magmatic profiles in the south, first volcanics are observed at 4.2s TWTT or deeper."

**Technical Corrections**

**Figures**

**Figure 1 :** The figure overall looks nice but some improvements are possible. For me there is a confusion between the legend and the caption and it is not clear what are the SDR (should be grey but I am confused with the colour of the belts), the Belts, the Basement from Stica (is it the same as the cratons?). For clarity, you might think of removing the Belts which are barely discussed in the paper. I'd also remove the Rio Grande Arch and the Torres Syncline which are not discussed at all in the paper. Also, it would be nice to indicate the Perolas margin (trivial for you but maybe not for everybody).

**Figure 2 :** The caption for a) and b) seems to not match the figure (maybe not the correct version). a) only show profile S1 and b) only S3. Both a) and b) show surfaces + units.

**Figure 4:** Panel b): I found a little confusing to have this black triangle on the graph. Wouldn't it be better to have it on the axis (e.g. "Max vertical SDR TWTT thickness (s)" and "Post-rift sediment thickness TWTT at max SDR thickness (s)")? Just a suggestion. Captions: maybe "at the same horizontal distance on the profile" sounds better than "at the same location". I had to scratch my head to figure out what it meant. As you want.

**Figure 7:** "Figure 7" is written twice (Nice figure BTW).

**Figure 8:** What is this vertical line, small horizontal line and small circle on panel b? It seems that the black dot represents the TWTT of the first proximal SDR but what are these other symbols? This is

nowhere explained it seems and I cannot figure it out. On panel c) and d) it would be nice to add a vertical axis with TWTT as this is the core of what you want to show with this figure. (Ok, you have a scale, but it's not easy just with this scale to know where is 4s or 6s.

**Text**

**Lines 53-56:** Could be nice to have at least one reference for these Feliciano Belt and pre-rift geology if any reader want to know more on this topic.

**Line 88:** "top basement remains parallel" parallel to what? To Moho I imagine.

**Line 104:** "at approximately 30 km": maybe good to say 30km from what (from eastern part of the line? From coastline?) or to remove it altogether as it is not the point of the sentence.

**Line 130-131:** You might add a "mainly" (sediment supply being *mainly* controlled by factors external to margin formation) as margin formation can also influence drainage system and thus also sediment supply. (Sorry, I'm picky but I like the topic :-)

**Lines 145 ff.:** I was confused with the lithospheric thermal re-equilibration. I struggled a long moment to understand how you integrated it until I realized you probably did not include it as you want to know "the bathymetry that would exist at present if no post-rift sedimentation had occurred." And post-rift sedimentation barely has an influence on thermal re-equilibration. Maybe it's not bad for ignorant people like me to mention somewhere how you handled it. What do you think? Or did I misunderstand something?

**Line 167:** "subduction dynamic subduction": One subduction too much and a missing subsidence.

**Line 178:** " The Austral segment of the South Atlantic margin of South American" a word missing

**Lines 204 and 206:** Where is Rio Grand do Sul? It seems to be nowhere on your maps. Probably good to include it on fig. 1.

**Line 247:** Chennin with one "n"

**References:**

- Rossetti et al. not in alphabetical order.
- Warner (1987) missing.
- Maybe not bad to check for other mistakes in the references.

---

## Author Comment (AC1)

*We thank Valentin Rimes for his detailed review. We have found the review very constructive and helpful for improving our paper. We have responded to all comments and suggestions and our responses are shown below.*

**Review 1 – by Valentin Rimes**

Dear authors, dear editor,

Thank you for this nice manuscript!

Overall, the manuscript is novel, of clear scientific significance, useful for the community and well argued. The aim, the focus of the paper and the conclusions are clear. It is overall well written, and the figures are clear (and look nice!).

Nevertheless, there is still room for some improvement to improve the readability and strengthen the argumentation. Precisions or corrections are definitely needed for the 4s vs 6s TWTT of first volcanics and for the generation of the results of the model in time domain (see details in the PDF). Some interpretations might be restructured. Some precisions seem to be needed for some methods. There are several potential small improvements to the figures and the text. Finally, I also propose a series of comments that are more proposition than corrections and that the authors might elect not to follow.

After these improvements/precisions/corrections, I'm sure that the manuscript will be a great contribution to the understanding of passive margins and a great addition to Solid Earth.

You'll find my detailed review in the supplementary PDF. Do not hesitate to contact me if something I wrote is unclear or if you have questions about my review.

Best regards,

Valentin Rime

**General Comments**

In this manuscript, Cassel et al. present seismic data from the South Atlantic and discuss the alongstrike variability of SDR and its correlation and influence on accommodation space. It is then shown that these results are consistent with variation in decompression melting during breakup.

Overall, the manuscript is novel, of clear scientific significance, useful for the community and well argued. The aim, the focus of the paper and the conclusions are clear. It is overall well written, and the figures are clear (and look nice!).

Nevertheless, there is still room for some improvement to improve the readability and strengthen the argumentation. Precisions or corrections are definitely needed for the 4s vs 6s TWTT of first volcanics and for the generation of the results of the model in time domain (see details below). Some interpretations might be restructured. Some precisions seem to be needed for some methods. There are several potential small improvements to the figures and the text. Finally, I also propose a series of comments that are more proposition than corrections and that the authors might elect not to follow.

After these improvements/precisions/corrections, I'm sure that the manuscript will be a great contribution to the understanding of passive margins and a great addition to Solid Earth.

**Specific Comments**

**Lines 134-135:** I'd say that these velocity values (for sediments) should be justified. A simple reference or a small sentence might be sufficient but at the moment they come out of nowhere. Or are they also coming from McDermott et al. 2019 like for the SDR?

Regarding post-breakup sediments, in line 135 we state that we use a k value of 0.4 km/s/km for interval seismic velocity dependence on depth. Figure McDermott et al. 2019 shows a k value nearer 0.5 km/s/s, however this value is likely to be too large for the thicker post-breakup sediment of the profiles to the south. We will also test using k=0.5km/s/s/ in the depth conversion and the follow-on flexural backstripping to produce water loaded accommodation space and present the results as a sensitivity test either on a figure in the main text or in a supplement.

**Lines 136-137:** "The SDRs of the Torres High profile are very thick and are most likely composed of basaltic flows. As a consequence, we use a higher velocity of 6.5 km/s for depth conversion." Did you use 6.5 km/s for all SDR of the study or only for the Torres High profile? From the phrasing, it is not clear to me. If other velocities where used, this should be stated. If the same velocity was used, you might also add a sentence to justify it as your observation "The SDRs of the Torres High profile are very thick" is only valid for the S1 profile. I agree that these changes have no influence on the conclusions of this paper but it would be scientifically nice to justify your choices (simplicity and comparability might be valid reasons on this one :-) ).

For simplicity we used 6.5 km/s interval seismic velocity for depth converting SDRs for all profiles (S1, S2, S3, S4).  McDermott et al. (2019)  show a laterally variable "skin" of lower interval seismic velocity about 2 km thick above deeper SDRs with 6.5 km/s. The average interval velocity for the whole SDR pile is likely to be slightly less than 6.5 km/s. We will test using a lower SDR interval velocity of 6.0 km/s in the depth conversion and the follow-on flexural backstripping to produce water-loaded accommodation space and present the results as a sensitivity test either on a figure in the main text or in a supplement. Because we only backstrip the post-breakup sediments (not the SDRS) this will have minimal influence on the determined water-loaded post-breakup accommodation space. This aspect of the discussion highlights why we focus in figures 4 and 8 on measurements in TTWT – the primary seismic reflection observation is in TWTT while a depth-conversion is a model with often substantial uncertainty.

**Lines 234 ff:** I find that this part on the Rio Grande Cone is not the strongest of your manuscript. I'd say that the Rio Grande Cone is mainly there because a large river brought sediments there. And then the high accommodation space caused by the not-so-magmatic margin allowed it to deposit on top of the passive margin. But let's make a thought experiment. What if this river brought sediments over S1 (or S2) profile? The same delta would probably also have deposited, but just further offshore, on top of oceanic crust or the very distal passive margin. So I'd say that the main control on the presence of this delta is mainly the presence of the river and the high accommodation space allowed it to deposit not too far offshore. The structure of the margin (magma very rich or not so rich) does control the accommodation space on the margin but not further oceanwards where there is anyway plenty of space (except exactly on the Walwis ridge but that's not a simple SDR). Maybe you could rephrase it as "the high accommodation space allowed the Rio Grande cone to deposit large thicknesses of sediments

on top of the margin". Anyway, I feel that this discussion on the Rio Grande Cone is not super strong or the most interesting of your paper and you might want to remove it from the abstract and/or summary (but not from the main text). What do you think?

In line 130 we remind readers that margin sediment thickness is dependent not just on accommodation space but also on sediment supply. Sediment supply is of course controlled by many external factors and varies substantially along margin (as shown for the Pelotas margin example). The purpose of flexurally backstripping the post-breakup sediments to give water-loaded accommodation space is to remove the consequences of laterally varying sediment supply. This enables the water-loaded accommodation space of the Torres High line (S1) to be directly compared with the Rio Grande Cone lline (S3). Perhaps we need to explain more clearly the purpose of flexural backstripping.

**Line 243:** "for the magma-normal margin profiles in the south, first proximal SDRs occur at 6s or deeper". That is for me the main problem of the manuscript. Your figure 3 and 8 show that it occurs around 4s and not 6s. This is correctly mentioned in the Abstract "first volcanics are observed at 4.2s TWTT or deeper" but not here. This also has implications for the consistency of your model (section 5.3). And for your conclusion ("In the time domain, a magma-rich margin, with sub-aerial SDR flows, shows first volcanics at ~2s TWTT while a "normal" magmatic margin has first volcanics at 6 -7s TWTT."). It's not a huge deal as it doesn't change the conclusions of your paper, it's still deeper than 2s. But it's not 6s, that's incorrect.

We agree totally with this comment and we need to correct this. In figure 8b we show the TWTT for first proximal SDRs with uncertainty. The solid circle should be in the centre of the range – and perhaps the horizontal arrows should identify both maximum and minimum values (not just the minimum value). The median value for line S3 and S4 then becomes slightly greater than 5s (not 4.2s). Closer inspection of figure 9e shows that the very first volcanics occurs not at 6s but at about 5.5 s., more comparable with the observation.

We would not expect the model prediction and observation to agree exactly; the model is a very simple one. We should also mention that the model prediction assumes a fully thermally equilibrated lithosphere while the Early Cretaceous Pelotas margin is not fully equilibrated and is still thermally subsiding which will slightly decrease the TWTT of first volcanics.

**Section 5.3 (lines 241-281):** I'm not 100% comfortable with this section. Maybe some things have to be restructured/rewritten or better explained. I see 4 problems/improvement potentials with this section (and maybe with the structure of the whole discussion). Here they are and I'll go more in details on each after.

1.      Methodology: I find strange to find methodological description here, I don't understand one part of the method and I'd argue that you need to include sediments to compare it with a real margin.
2.      The model shows 6-7s TWTT while the seismic 4s.
3.      I don't understand why the results of the modelling come only to support the TWTT of first volcanics and not to support also sections 5.1 and 5.2
4.      I don't understand why you focus only on TWTT of first volcanics to identify magmarich/magma-normal margins.

See detailed replies to comments 1-4 below.

Ok, let's go in the details of each comment.

**Comment 1:**

Lines 246 – 263 and 275-278: I have the feeling that this is more methods and should not be placed in the discussion chapter (5). I would make a new section before the discussion to present the methods. As a lazy reader, I often want to read the discussion chapter without the details of the method and I thrust the (lazy) reviewers to have checked the methods :-) . At the moment, I think it "dilutes" a bit your discussion points. What do you think? Described in

Our paper is primarily an observational paper. Its purpose is not to test a model – it is to make observations from data.

We use the simple model (line 246 and onwards) only for the purpose of trying to understand our observations. We therefore place the simple model in the discussion – indeed we place it at the end of the discussion.

Line 275 ff.: Here I don't get how you calculated your sections in time. Ok, you assumed that the Moho is at 10s (BTW on the figure it's a bit deeper than 10s) but how did you calculate the other reflectors? Did you use the velocities mentioned in section 4 and you used this 10s-rule just to compensate for the lack of good constrains on the velocity of the crustal basement or…? This needs to be better explained.

We need to explain more clearly the use of Warner's 10 second rule for Moho TWTT.  Wie will revise the text to do this. Warner (1987) observes and explains why the Moho TWTT for thermally equilibrated lithosphere is always close to 10s irrespective of crustal basement thickness and sediment thickness above. It is an approximation but a useful rule. It means that if the Moho is at approximately 10s and we know basement thickness, then the TWTT to top basement can be calculated (we assume a basement seismic velocity of 6.5km/s). This bottom approach means, for thermally equilibrated lithosphere,that the TWTT of top basement to first order is independent of sediment thickness. The reasons for this (explained by Warner  1987) are a combination of isostasy and the relationship between density and seismic velocity.

Also, how can you model a section in time and compare it with your seismic if you don't model post-rift sediments? I understand that the post-breakup thermally equilibrated sections (fig. 9 c and d) are just a concept and not used to compare with reality 1:1 and thus it's not a big problem not to model sediments. But you compare the time-converted sections (fig.9e, f) 1:1 with real-world examples. And I'd argue that the presence of sediments instead of water would have a big influence as the seismic velocities are completely different. Without a change, I'd say that the comparison is invalid. But as I did not fully understand your methodology, maybe I missed something.

See earlier comment on the use of Warner's 10s rule..

As they are four problems with this time-converted section (two highlighted here, one in comment 2 and one in comment 4), you might elect, in an extreme case if you can't correct it, to just remove this time comparison and only use the modelling to show the change in volume of volcanics and changes in accommodation space (see also comment 3 and 4). The paper would still be strong, relevant and useful without this. But of course, it's nice to have.

The paper is primarily an observational paper so, yes, the simple model at the end of the discussion could be omitted. However we believe that the simple model forms a useful part of the discussion of the observations.

**Comment 2:**

The model shows first volcanic material at 6-7s TWTT while the seismic shows it at 4s (as already discussed above). Maybe you have to re-run your model with decompression melting starting a bit

earlier to match with your observations at 4s. Or what if you include sediments in your time-migrated model (as discussed in comment 1)? As they have velocities higher than water, that might pull up the appearance of the first volcanics and better fit your observations.

See earlier comments. We will revise the text to address this.

**Comment 3:**

I don't understand why your modelling results only come at this point of the manuscript.

See response earlier. The aim of our paper is to make observations – the model is only used to try to understand the observations – the purpose of the paper is not to test the model.

In the main text, you only use it to justify your 4/6s TWTT but in the abstract and the summary, you also use it to justify the difference in SDR thickness and accommodation space ("*The observed inverse relationship between post-breakup accommodation space and SDR thickness is consistent with predictions by a simple isostatic model of continental lithosphere thinning and decompression melting during breakup*" and "*The observed inverse relationship between postbreakup accommodation space and SDR thickness is predicted by a simple isostatic model of continental lithosphere thinning and decompression melting during breakup*").

I think you have to discuss it a little bit more in the text. You need to justify it in the text before presenting them as "conclusions". At the moment you only have one sentence "lost" in section 5.3 about accommodation space (line 271-273) but nothing about volume of volcanics. Of course, I agree with you, your model show it, but you should mention it clearly in the text. Why not use them in the text to justify your point 5.1 and 5.2? Your model indeed does not only show the difference in TWTT of first volcanics, but also that the thickness of volcanic material and accommodation space changes with the different parameters. Why only use it for 5.3, right at the end?

We understand this comment and we will revise the text to explain this more clearly.

Your order of argumentation at the moment also makes less sense as you already concluded in section 5.1 that the margin is more magma-rich in the N. Why would you then test the same hypothesis with a model in section 5.3 if you already came to this conclusion? I see two options to circumvent this problem: either you present your modelling first (as already proposed on comment 1 and see further comments) and then can also use these results to support your point 5.1 and 5.2. Or you say that you don't need to prove this with a model and that the thickness is the volcanic sequences is enough to prove we are more magmatic (and I would agree with this). If so, you wouldn't need the modelling at all and you can discuss the TWTT of first volcanics based on real-world data alone (but I'd say that your model is still a good addition to the discussion). Maybe you see another solution to circumvent the problem?

See earlier replies about what we see the role of the modelling to be in the paper. We see this paper a an observational paper, not a modelling paper. We include the modelling in the discussion (at the end) only to help understand the observations.

**Comment 4**

I see another problem in the summary linked to section 5.3: "Our study shows that SDRs are not synonymous of magma-rich margins; the TWTT of first volcanics may provide a better approach to distinguishing magma-rich margins from margins with normal magmatic addition". Again, this has not clearly been discussed before coming to the summary. Your whole study (observation + model) showed

that the thickness of volcanic material [and accommodation space as a consequence] also distinguishes magma-rich and magma-normal margins (and not only TWTT of first volcanics). Why do you only focus on the TWTT of first volcanics to determine whether it's a magma-rich or magma-normal margin in the summary and abstract? Is it because it's easier to measure than thickness of volcanics and accommodation space? That would be a good reason but should be discussed in the text.

We understand reviewer 1's comment and you identify an important point. We will increase the explanation and discussion of the observation of TWTT of first volcanics and its correlation with volcanic volumes. This is perhaps the most important result of the paper and at present it is perhaps slightly hidden.

Also, the age of the margin might also play a role here. Do you think that this boundary would also be at 4s TWTT on a very recent margin with almost no post-rift sediments yet and less thermal subsidence? I'd say no (although I do agree that after a while, thermal subsidence is close to 0 and the accommodation space above the first volcanics is anyway filled (as on your example)). In the case of a very young margin, the thickness of magmatic material would be the best parameter to determine between magma-rich and magma-normal margins. This brings us back to the previous paragraph: why is the TWTT of first volcanics a better method that other methods? This limitation should probably be discussed in the text.

Reviewer 1 raises an important point. The age of a margin will affect the TWTT of first volcanics because a young margin will not be thermally equilibrated. We will explain this effect in the revise text.

**Suggestion of improvement for the structure of the discussion:**

As seen from the comments above, the reasoning path of the discussion section could be changed to something like:

We see several differences across the strike of the margin: thickness of volcanics, accommodation space and TWTT of first volcanism (2 vs. 4/6s) ❼ We relate this to a change in the timing and volumes of magmatic production ❼ Our model allow to test this rather we think the model allows us to explain the observation ❼ Thickness of volcanics, accommodation space and TWTT of first volcanics of the model fit real-world observations (story of 6 vs. 4s and inclusion of sediments in the model apart) This is shown in the Chenin et al 2023 paper – we need to expand the text to explain this in more detail ❼ These differences in thickness, accommodation space and TWTT of first volcanics can be explained by a different volume and appearance time (or beta factor) of decompression melting (along with the other arguments you already give in section 5.1) as is shown in the Chenin et al 2023 paper ❼ Both thickness of volcanics, accommodation time and TWTT of first volcanics provides a way to determine magma-rich margin more reliably than the presence of SDR. Yes! ❼ Because TWTT of first volcanics is easier to measure, it probably provides the simplest way to determine magma-rich margin (or another argument as to why this parameter is important) Yes! ❼ then a last section of the discussion with current section 5.2 about accommodation space (including that it is also confirmed by the model). We will increase the discussion of this in the text

That seems like a big change but basically you can just copy and paste most of your existing text. But that's only just a suggestion to use your modelling to support all your points and make your discussion clearer, more relevant and more impactful. You might elect to not follow it at all, that's not a problem.

We understand reviewer 1's comments above and will modify the text.

**Summary (lines 282-298):** As discussed before, you might consider removing the Rio Grande Cone story as it not the most important outcome of the study. Anyway, I'd maybe put the point "The observed inverse relationship between post-breakup accommodation…" right after the point "Post-breakup accommodation space correlates inversely with SDR thickness…" as those are linked.

See earlier response to this – the flexural backstripping corrects for the variable sediment supply along the margin

Line 296, this 6-7s has to be clarified. 6-7s is only from the model at the moment, not from seismic data.

See earlier responses to this point. Reviewer 1 is correct and we need to amend the text and figure to be consistent.

Line 297-298: In light of the discussion above, the sentence could be slightly changed to "may provide the simplest approach to distinguishing …"

Good point – we agree and will amend.

**Abstract:**

Maybe linked with comment 3 above, I'd also maybe reshuffle the order of the sentences of the abstract to put your model upfront [but this is just a suggestion]:

As discussed earlier, we see the paper as an observational paper rather than a modelling paper. We use the model only to try to explain the observations.

"We show that post-breakup accommodation space correlates inversely with SDR thickness, being less for magma-rich margins and more for magma normal/intermediate margins. *The observed inverse relationship between post-breakup accommodation space and SDR thickness is consistent with predictions by a simple isostatic model of continental lithosphere thinning and decompression melting during breakup.* [The Rio Grande Cone, with large sediment thickness, is underlain by small SDR thicknesses allowing large post-breakup accommodation space.] A relationship is observed between the amount of volcanic material and the TWTT of first volcanics; first volcanics are observed at 1.25s TWTT for the highly magmatic Torres High profile while, in contrast, for the normally magmatic profiles in the south, first volcanics are observed at 4.2s TWTT or deeper."

**Technical Corrections**

**Figures**

**Figure 1 :** The figure overall looks nice but some improvements are possible. For me there is a confusion between the legend and the caption and it is not clear what are the SDR (should be grey but I am confused with the colour of the belts), the Belts, the Basement from Stica (is it the same as the cratons?). For clarity, you might think of removing the Belts which are barely discussed in the paper. I'd also remove the Rio Grande Arch and the Torres Syncline which are not discussed at all in the paper. Also, it would be nice to indicate the Perolas margin (trivial for you but maybe not for everybody).

We agree - reviewer 1 makes several good suggestions here – we will revise the figure following that advice to make it clearer.

**Figure 2 :** The caption for a) and b) seems to not match the figure (maybe not the correct version). a) only show profile S1 and b) only S3. Both a) and b) show surfaces + units.

Agreed – we will correct the caption text

**Figure 4:** Panel b): I found a little confusing to have this black triangle on the graph. Wouldn't it be better to have it on the axis (e.g. "Max vertical SDR TWTT thickness (s)" and "Post-rift sediment thickness TWTT at max SDR thickness (s)")? Just a suggestion. Captions: maybe "at the same horizontal distance on the profile" sounds better than "at the same location". I had to scratch my head to figure out what it meant. As you want.

Good point – we agree. We will amend the figure.

**Figure 7:** "Figure 7" is written twice (Nice figure BTW).

Error noted – thanks – we will correct

**Figure 8:** What is this vertical line, small horizontal line and small circle on panel b? It seems that the black dot represents the TWTT of the first proximal SDR but what are these other symbols? This is nowhere explained it seems and I cannot figure it out.

We agree – this is not adequately explained and is confusing – the vertical lines show the range of uncertainty. We will amend figure 8b and improve text.

On panel c) and d) it would be nice to add a vertical axis with TWTT as this is the core of what you want to show with this figure. (Ok, you have a scale, but it's not easy just with this scale to know where is 4s or 6s.

Understood – but the owner of the seismic requires this presentation format of TWTT scale.

**Text**

**Lines 53-56:** Could be nice to have at least one reference for these Feliciano Belt and pre-rift geology if any reader want to know more on this topic.

Understood – will add reference

**Line 88:** "top basement remains parallel" parallel to what? To Moho I imagine.

Understood – we will explain this more clearly

**Line 104:** "at approximately 30 km": maybe good to say 30km from what (from eastern part of the line? From coastline?) or to remove it altogether as it is not the point of the sentence.

Understood – we will amend text

**Line 130-131:** You might add a "mainly" (sediment supply being *mainly* controlled by factors external to margin formation) as margin formation can also influence drainage system and thus also sediment supply. (Sorry, I'm picky but I like the topic :-)

Understood – point taken – will amend text

**Lines 145 ff.:** I was confused with the lithospheric thermal re-equilibration. I struggled a long moment to understand how you integrated it until I realized you probably did not include it as you want to know "the bathymetry that would exist at present if no post-rift sedimentation had occurred." And post-rift sedimentation barely has an influence on thermal re-equilibration. Maybe it's not bad for ignorant people like me to mention somewhere how you handled it. What do you think? Or did I misunderstand something?

Reviewer 1's understanding  is correct – we are not rewinding post-breakup thermal subsidence. We will add text to make this clearer.

**Line 167:** "subduction dynamic subduction": One subduction too much and a missing subsidence.

Well spotted – we will correct

**Line 178:** " The Austral segment of the South Atlantic margin of South American" a word missing

Understood – should read "The Austral segment of the South Atlantic margin of South America" – we will correct

**Lines 204 and 206:** Where is Rio Grand do Sul? It seems to be nowhere on your maps. Probably good to include it on fig. 1.

Ah – Rio Grande do Sul is the most southern state of Brazil (and where the first author comes from) We will amend the text to make this clearer..

**Line 247:** Chennin with one "n"

Thanks - we will correct

**References:**

- Rossetti et al. not in alphabetical order.
- Warner (1987) missing.
- Maybe not bad to check for other mistakes in the references.

We will add and correct references

---

## Author Comment (AC2)

*We thank reviewer 2 for the review. We have responded to all comments and suggestions which we found constructive and will improve our paper. Our responses to the reviewer's comments and suggestions are shown below.*

**Review 2 - Anonymous**

Review of "Along strike variations of volcanic additional controlling post breakup sedimentary infill"

This manuscript provides a useful insight into the ongoing evolution of magmatic margin and their associated sedimentary basins. Although the focus is the Pelotas margin, there are wider implications.

There are number of details comments below, but there are three specific aspects that need to be considered before the manuscript is appropriate for publication.

- Throughout the study there is no clear differentiation of basement and no discrimination between oceanic and continental crust. For margin development I think it is critical that more consideration is put into this component. While the data itself may not be present to define this, there are sufficient publications on similar data that more detail could be included. I also think more coverage of SDR formation could be included. This is important because it is fundamental to how we interpret such margins and especially as SDRS are an intrinsic part of margin formation. The way the manuscript comes across it appears that SDRs are emplaced onto 'basement', rather than the emplacement of SDRS (or at least outer SDRs) are simply the shallower component of an over thickened magmatic/oceanic crust development. Whether the authors agree or disagree with this it is essential that this argument is presented. This is essential because 1) where SDRs are limited is this just because there is greater proportion of sub-SDR magmatism hence along trend magma volumes are not simply a function of SDR thickness. 2) When rotation is undertaken magmatic crust thickness is not simply a function of present day vertical thickness – what impact does this have on estimations of magma supply?

Reviewer 2 identifies an important point. It looks strange and non-geological to show sections where oceanic crust and continental crust are undifferentiated (although with different ages) but the SDRs have a different colour suggesting that the SDRs are deposited over a crust that already formed. Ideally we should differentiate continental basement, magmatic additions (SDR/Underplates) and oceanic crust – but this is not possible to do reliably with the available data. It is also not necessary because we do not discuss how the oceanic crust and magmatic addition form. We will modify figures 2, 5 and 8.

Our aim is to identify how seismic observations in TWTT can be used to identify magma rich margins without involving the difficult and unreliable interpretation of basement type (thinned continental crust, hybrid crust, oceanic crust). We aim to focus on observations that can be reliably determined.

• Current SDRs models invoke emplacement at sealevel; the backstripping suggests significant water depth when backstripped. This is incompatible with SDR emplacement but is not discussed. A more comprehensive discussion of this has to be included as to how appropriate is backstripping for such margin and what the role of isostacy and thermal perturbations are when backstripping is undertaken.

The purpose of our flexural backstripping is to produce sediment post-breakup accommodation space. We do not restore post-breakup thermal subsidence to produce breakup palaeobathymetry. Our aim is not to examine the deposition of SDRS and their palaeobathymetry.

• A central premise of the manuscript is the control of the margin on overburden and along margin variation– as noted in (1) a much more compressive discussion of crustal thickness (continental vs oceanic) has to be undertaken including what sits beneath SDRs. In addition there is no discussion of the role of paleogeography and sediment supply, eg. Drainage and sediment entry in any substantial way. There is a discussion that the backstripping sediment can be used to estimate thickness but this needs to be correlated with additional sediment supply data.

The aim of the paper is not to explain the evolution of the margin but is to analyse, using quantification methods, the relationship between the magnitude of volcanic additions and post-breakup accommodation space for sediments. We do not need to examine the sedimentary-evolution of the margin so it is not important where the sediments come from. Also we do not need to examine the geological evolution of the margin sections. We deliberately avoid speculating on the nature of the underlying crust which cannot be determined with any certainty.

• Suggest there should be consistency of using depth converted data Currently starts with TWT then Z(m) – suggest more consistency having depth conversion from the outset

For good reason we prefer to use the primary seismic observation which is in TWTT. Depth sections are models which are dependent on the seismic velocity used in the depth conversion. At volcanic margins it is very difficult to accurately determine seismic velocities for depth conversion and any resulting depth sections are uncertain.

Specific comments

Introduction – this really needs more on SDR and magmatic margin formation processes that addresses crustal nature below SDRs – need to define what basement is at each part of the section,

The formation processes of SDRs and magmatic margins is not the aim of the paper. Our observations do not depend on the formation processes.

Regarding basement, we will update figures and text as said earlier.

This is specifically relevant for Line 44 – it is the volume of magma that is important not just SDRs.

We purposefully avoid estimating the total volume of magma including intrusives – it is difficult/impossible to determine with any accuracy as explained in Tugend et al. (2018) and Chenin et al. (2023)). We focus on the volcanic/extrusive section, since it can be mapped/observed. We assume that the thickness (and interval TWTT) of the volcanic/extrusive section is a proxy for the

total magmatic addition since it represents a fraction of the total magmatic system, usually considered to be 1 to 2 or 1 to 3 (Crisp, J.A., 1984.)

Line 45 – how representative is this and what implications are appropriate for other margins?

We will reword this to better explain what we mean (and not overstate)

Line 85 – basement characterisation in text and associated figures is required

We will update the figures and text regarding basement as discussed earlier. Our aims are to determine the relationship between the magnitude of volcanic additions and post-breakup accommodation space for sediments which does not need the nature of basement to be known.

Line 88 – "top basement is a smooth horizon onto which the SDRs downlap" this is not substantiated by the data presented and is very different interpretation than current models of SDR formation.

The base of SDRs is difficult to map and we will state that in the text. We make the assumption that top basement is smooth as an approximation so that we can make measurements.

Line 91-2 – 1$^{st}$ mention of basement shape – tapering, box shape. No discussion of what this means, how it is defined and what relevance is. Also how much does this geometric configuration rely upon definition of continental or oceanic crust?

We understand reviewer 2's comment and need to explain "basement shapes" more clearly which we will do in the revised text. We will also add a reference to Chenin et al. (2023) which expands on factors controlling margin shape. Because our paper aims to make quantified observations, we need to make compromises such as using simple first order interpretations rather than complex interpretations that may not substantially change the quantification results.

Line 95 – this seems slightly unnecessary for this paper

We agree – this is not necessary and we will delete

Line 100 Figure 2 caption is incorrect

We agree – we will correct

~Line 105 – suggest this is all depth converted from the outset

As explained earlier we prefer to work with time sections and TWTT which is the primary seismic observation rather than depth sections which are a model dependent on depth conversion seismic velocity which is difficult to determine accurately at volcanic margins.

Figure 3 – this highlights the importance of defining basement and a more detailed discussions SDR and its sub-crust. For example in b if SDR and magmatic crust is considered as a single crustal type (ie overthickened oceanic crust) this has a very different implication.

Ideally, we agree that it would be good to define basement type distribution including magmatic underplate and the limit of continental crust and the onset of oceanic crust. However, these are not easy to observe with accuracy and would produce non-unique interpretations. In our paper we take the SDR section as a mappable unit and assume that it is representative of the "total" magmatic addition and that basement can be treated, independent of its composition/history, uniformly. Fig. 8 demonstrates that these simplifications are reasonable since they predict the TWTT of the onlap of magmatic systems.

Line 124 – very difficult to make a discussion of sediment supply with more extensive discussion, and indeed from a 1D perspective rather than 2D at least.

The purpose of figure 4a is to show the difference between the 4 profiles. We agree that the vertical axis values will depend on sediment supply which is unknown. This is the reason that we determine accommodation space using flexural backstripping which corrects for the laterally variable unknown sediment supply.

Line 129- volcanic addition would be fine if it was all onto continental crust but as defined basement includes oceanic crust therefore need further discussion of SDR formation

Yes we agree. We will add text to explain this. This is also why the TWTT of first volcanics is so useful.

Line 134 – where is crustal basement, how would this change?

The post-rift accommodation space would not change. We will clarify.

Figure 5 – generally assumed that SDRs are sub aerial but backstripping does not restore these the SDR packages to sea level. This must be discussed further

See earlier response to this comment – we do not restore post-breakup thermal subsidence

Line 145 – if accommodation space is estimated from backstripping, then need to consider the point raised above – i.e. SDRs at sea level. This comment applies to this whole section.

See earlier response to this comment – we do not restore post-breakup thermal subsidence

Line 165-170 – there are some significant assumptions made in this statement. This needs further justification.

We will explain this more clearly

Line 188- magmatic volume - needs further justification as to whether extrusive SDR is equivalent to magmatic volume which again goes back to defining basement.

We will add further discussion

Line 228-233 – what is the evidence of significantly underfilled sun-rift basins? This would be apparent from the geometry of the basin fill

The point we make here is that if syn-rift accommodation space is filled with volcanics it is not available for later (post-breakup) accommodation space.

Line 240 – not sure how this fits with the original premise of the paper?

We believe that this is a very important observation with important implications.

---

## Author Response (AR1)

Dear Mohamed,

**Manuscript Revision - Along-strike variation of volcanic addition controlling post breakup sedimentary infill: Pelotas margin, Austral South Atlantic" by Cassel et al.**

We have revised our manuscript in response to reviewers comments and have loaded it onto the Solid Earth website. We have responded to all reviewers comments and we believe that the revised manuscript has greatly benefited from the reviewers' comments. The track changes version of the revised manuscript shows our revisions. In some cases our comments in the right-hand margin of the track-changes document explain where we believe that the reviewers' comments or suggestions are incorrect.

Below, we briefly summarise our responses to the main comments made by the two reviewers.

- R#1 (reviewer 1) suggest that we should restructure the paper with a primary focus on modelling. We very much disagree and see our paper as being primarily observational. The purpose of the simple model that we use at the end of the discussion is to attempt to understand our observations. Our intention is not to use our observations to test a mode.

- R#1 suggest that we omit the analysis of the Rio Grande do Su Cone profile S3 because that profile has very large thicknesses of sediment. This suggestion misses the whole point of why we flexurally backstrip the 4 sections to determine water loaded post-rift accommodation space. We have explained the purpose of flexural backstripping in more detail to hopefully remedy this. We also emphasise that its aim is to determine water loaded post-rift accommodation space and not to produce a restored cross-section at base post-rift (we intentionally do not reverse model post-rift thermal subsidence).

- R#1 correctly identified inconsistencies in our reporting of observed TWTT of first volcanics. We have revised text and figures to be consistent.

- R#1 requests that we justify the seismic velocities that we use for depth conversion of sediments and SDRs. We have done this. We also explain that SDR seismic velocity has no impact on our results since we do not flexurally backstrip the SDRs to determine post-rift accommodation space.

- R#2 (reviwer2) proposes that we work with depth converted cross-sections and make our observations in depth not TWTT. We disagree and explain that, because of

substantial uncertainties in the seismic velocities required for depth conversion, depth sections are very inaccurate. We have explained more clearly why we prefer to work with time domain seismic section, TWTT being the primary observation and not dependent on the seismic velocity model.

- R#2 requests that we provide more information about the SDR formation processes and the nature of the basement onto which they are deposited. We explain more clearly that our observational strategy does not require these to be known and that we deliberately avoid speculative interpretation of SDR formation process and basement type.

- R#2 request that we also quantify intrusive magmatic addition. We explain that such quantifications are extremely inaccurate and ambiguous and that we deliberately avoid doing this.

Regards,

Marlise

---

## Referee Report (RR1)

Overall, the authors greatly improved the manuscript and clarified most of the points that were unclear in the previous version. As already stated, the paper is novel, very interesting and relevant for the geological community.

I still have a (very) few minor comments and one serious concern about the data (or at least fig. 3), not supporting the conclusions. This is an important concern and should it be clarified.

**Figure 8:**

1. Missing a), b), c), d) labels on the figure

**Section 5.3:**

2. *In contrast, for the magma-normal margin profiles in the south, first proximal SDRs occur at **6 s or deeper** (Fig. 8c).*

Sorry, this was a concern of my first review and this is still not resolved now. I went in more detail looking at figure 3 and 8 and I don't understand how you come to the results of figure 8b. The data on figure 3 show very different values. Or maybe I don't understand what you mean by first proximal SDR. For me I understand it as the first (shallowest) identification of rocks belonging to the SDR package. Or the point where the horizons "top of SDR" and "base of SDR" diverge. And if you look at this, then I cannot understand how your uncertainty can be that big. For S3, it encompasses half of the whole SDR package. If we look at figure 2a and 2b, the distinction between what is SDR and what is not part of it seems quite clear. Furthermore, TWTT data are raw data and there is therefore no (or very minimal) uncertainty. It's not like depth-converted data which can feature large uncertainty depending on the velocities assumed. And by having these huge uncertainties and using the mean of these, you end up considering TWTT values much higher than what your fig. 3 do show for the first proximal occurrence of SDR.

In the figure below, I show how I would interpret the TWTT of first proximal SDRs (black rectangles with values) and how you did interpret it (green rectangles with values, taken from fig. 8b on the right). And they show very different results for profiles S2, S3 and S4.

[Figure]

[Figure]

So either:

    a) I don't understand what you mean with "first proximal SDR". But then the reader might also not understand it and you would have to explain it better. I fail to see what it could mean else. I assume it does not mean the oldest flows (or reflectors) of your SDR package (which would cover a range of TWTT) instead of the first observation of SDR rocks (which

would be a point) as your uncertainty bracket of profile S3 encompasses several SRD packages (we can see it on fig. 2b).

b) The TWTT scale of figure 3 is not correct.

c) Your plot on fig. 8b is incorrect and should plot lower values with lower uncertainty.

d) I completely misunderstand something else. But then the reader might also misunderstand it and it should be clarified.

So for the moment, I would say that your data (fig. 3), do not support your conclusions, which is a major issue. This includes the mention of 5.5s TWTT in the abstract, in the Summary, as well as the 5.25 (not consistent BTW) on line 324. For me your data show 4.0 -4.5s TWTT and not 5-6s TWTT.

But your conclusions still hold true for a value of 4.0-4.5 s. This is still higher than 1-2 s TWTT for the magma-super-rich S1. But this has to be sorted out.

3. *In contrast the normal magmatic addition model shows (Fig. 9a) the upper surface of first volcanics at ~ 2 km water depth corresponding to submarine lava flows erupted onto thinned continental crust.*

It would not be bad to add a small comment saying that this suggests that SDRs seismic facies might not only form subaerially or near sea-level as presented in classical models. Which is not a surprise as they are based on very few samples. You mention this in your answer to the editor but I think it should also be mentioned in the paper as it is an important observation.

4. *Invoking Warner's 10 s rule for the Moho TWTT for thermally equilibrated lithosphere allows the cross-sections shown in Fig. 9c and Fig. 9d to be converted into the time domain as shown in Fig. 9e and Fig. 9f. To do this basement thickness **is converted to interval TWTT** and subtracted from 10s to give the TWTT of top basement; this estimate of TWTT of top basement is therefore independent of the interval TWTT of bathymetry and post-rift sediments.*

You have to give the velocities you used. In your answer to my first review, you mentioned that you "assume a basement seismic velocity of 6.5km/s". Fine but this should also be mentioned in the paper itself, not just in the answers.

**Summary**

5.  *In the time domain, a magma-rich margin, with sub-aerial SDR flows, shows first volcanics at ~1 – 2s TWTT while a "normal" magmatic margin has first volcanics __at 5 – 6 s TWTT__*.

As mentioned above, this is not supported by your data (fig. 3) that rather show 4.0- 4.5s.

6.  *Our study shows that the TWTT of first volcanics may provide an alternative approach for distinguishing magma-rich margins from margins with normal magmatic addition compared to estimating total magmatic volumes*

As mentioned in my first review, I think that this should be introduced in section 5.3 and not only in the summary where it comes for the first time. I think you should add at least a few words in 5.3 to say what other approaches are used in the literature (you provide an alternative to what?) and in which aspects your alternative is interesting (obviously, it's easier to measure, even with seismic data of lower quality) and maybe what are the limitations of this alternative approach. Then it would be perfectly fine to include it in the Summary.

---

## Author Response (AR3)

**Editor's Comments dated 12 June 2024**

Dear Dr Cassel et al,

Thank you for the prompt reply to my comments.

I am not fully convinced with your response regarding the depositional environment of SDRs. Most direct observations of SDR-like features (i.e., from offshore drilling/dredging or from analogues exposed in the field) suggest that inboard ones are emplaced in subaerial or shallow water environment, while outboard ones might form in shallow marine setting (e.g., Direen and Grawford, 2003/https://doi.org/10.1144/0016-764903-010; and reference therein).

In any case, this is a very debated topic and since the additional arguments that you've provided will be publicly available to the readers, I am happy to see this latest version of the manuscript published.

I will inform the Support Team of my decision, which will be in touch regarding the manuscript production.

Congratulation for this nice piece of work and thank you for choosing to publish in our special issue.

Mohamed Gouiza

**Response by Authors to Editor's Comment of 12 June 2024**

We understand your concerns that many readers may only be familiar with the sub-set of SDRs that form at sea-level.

A fundamental problem may be what is meant by the term SDRs. The term is shorthand for volcanic seaward dipping reflectors (the volcanic is usually omitted) and sensu-stricto it is a purely descriptive term. The most investigated SDRs are of course the spectacular thick ones with long flow lengths which formed subaerially or near sea-level (e.g. Voering, Moere, Demerara Plateau, Pelotas). These SDRs come immediately to mind when one reads the word SDRs – but magmatic extrusives displaying SDR characteristics also form in deep marine environments particularly at margins with normal magmatic addition.

We have modified the text and added appropriate references to make this point clearer.

**Response to Editor's Comments of 4 June 2024**

Dear Cassel et al.,

Thank you for revising your manuscript and thoroughly replying to the reviewers' comments.

After reading the revised manuscript and your rebuttal, I think that there are two points that were raised by the reviewers that still require your attention:

1- The use of time (i.e., TWT) to describe primary observations from the seismic lines:

in your rebuttal you justify the use of time instead of depth by the lack of accurate/reliable seismic velocities, which would make any depth conversion highly uncertain.

However, your own modelling work (i.e., 2D flexural back-stripping) relies on depth converted sections.

This make me wonder how reliable are the modelling results that you have obtained?

- i.e., post-rift accommodation and its relationship with breakup volcanic addition.

Response

We apply flexural backstripping and decompaction to post-rift sediments only; we do not apply it to the extrusives volcanics below post-rift-sediments. As a consequence we avoid errors arising from the depth-conversion of extrusive volcanics. Errors in the depth conversion of post-rift sediment thickness will exist and affect the magnitude of water-loaded accommodation space determined by flexural backstripping and decompaction. However these errors are consistent between profiles and relatively minor so that the relative differences in accommodation space between profiles and our overall observation and interpretation are not changed.

2- One of your main conclusions is the depth in time at which first volcanics occur

- i.e., ~2s TWTT for magma-rich margin, and 6-7s TWT for "normal" magmatic margin.

This is confusing for two reasons:

(i)    as one of the reviewers rightly highlighted, current models of SDR emplacement assume that they are subaerial flows, as you mention your conclusions. If these depths are not emplacement depth then what are they exactly?

Response

Magmatic extrusives commonly take the form of seaward dipping reflectors (SDRs), resulting from the lateral migration and accretion of both subaerial and submarine lava flows during rifting, breakup and initial sea-floor spreading (Harkin et al., MPG 2020). The reviewer statement (i) above is incorrect ; it implies that the current model of SDR emplacement assumes that all SDRs are formed as sub-aerial flows. It is only correct to say that some SDRs, usually with long lateral flow lengths, are emplaced as subaerial flows. However this does not apply to all SDRs. Many occurrences of extrusive

magmatism at rifted margins (perhaps the majority) with sea-ward dipping attributes (i.e. SDRs) have deposition with relatively short flow lengths and are emplaced in relatively deep water.

(ii) What are the geological/physical processes/factors that control this depth of first volcanics?

Response

The water depth of first volcanics is controlled by the isostatic consequences of the relative timing of crustal/lithospheric thinning and the onset of melt production by decompression melting (see Chenin et al 2023). Factors advancing the initiation of melt production with respect to crustal/lithospheric thinning are elevated lithosphere and asthenosphere temperature and/or inherited lithosphere chemical enrichment. Factors delaying the initiation of melt production with respect to crustal/lithospheric thinning is inherited lithosphere chemical depletion.

I think that these aspects need to be addressed in the manuscript to remove some of the remaining ambiguities regarding your work and your results.

Mohamed Gouiza

---

## Author Response (AR4)

**Authors Response to Reviewer's Comments**

**Comment 1**

We have revised Figure 8 adding a, b, c and d labels.

**Comment 2**

We have updated figure 8b, c & d and associated text to make clearer our measurement and reporting of the TWTT of first proximal SDRs. Our revision explains more clear the uncertainty range shown in figure 8 and how it arises. The revised text is as follows:

"Examination of the seismic reflection sections in the time domain shows that the TWTT for the first appearance of proximal SDRs is also very variable. The observed north to south variation of TWTT of first volcanics is plotted as a function of latitude in Fig. 8b. For the purpose of showing the uncertainty in the measurement, two measurements of the TWTT of first proximal SDRs are plotted for each profile corresponding to lower and higher measured values. The term SDR *sensu stricto* simply means "seaward dipping reflector", and while the common use of the term is applied to volcanic seaward dipping reflectors, seaward dipping reflectors can also be generated by sedimentary sequences within fault controlled half-grabens. The lower TWTT values shown in Fig. 8b may correspond to either the onset of volcanics or sedimentary accumulations; their exact nature cannot be reliably determined using the available seismic data alone. In contrast the higher TWTT values shown in Fig. 8b represent a much more certain measurement for the onset of volcanics. The TWTT of first volcanic SDRs, using either lower or higher measured values, shows an inverse correlation with the magnitude of volcanic addition shown in Fig. 8a.

The lower and higher measurements of first proximal SDRs are shown in Fig. 8c for the magma-rich margin profile S1 over the Torres High in the north, and in Fig, 8d for the magma-normal margin profile S3 in the south. For profile S1, TWTT measurements lie in the range 1.2 to 2.2 s. These values are similar to those reported by Mutter et al. (1982) and Planke et al. (2000) for the long flow-length SDRs on the Voring segment of the Norwegian margin which formed at or above sea-level and have subsequently thermally subsided. For profile S3, TWTT measurements lie in the range 4.2 to 6.5 s similar to those reported by Planke et al. (2000) and Hinz et al. (1999) for the deep marine erupted SDRs on the Exmouth Plateau the Argentine margins."

**Comment 3**

We have expanded the text making it clearer that it has long been known that SDRs do not form exclusively subaerially and that they also form in deep-water. We cite references making this point (Hinz et al 1999, Planke et al. 2000). If there is a common misconception that SDRs only form

subaerially, then this additional text should help to correct that. The additional text making this point is as follows:

"SDRs with long flow lengths and large thicknesses, which form by extrusive magmatism in a sub-aerial environment, have been extensively studied (Mutter et al., 1982; Planke et al., 2000; McDermott et al., 2019; Harkin et al., 2019). However volcanic SDRs also form in a deep marine environment as voluminous effusive sheet flows (Planke et al., 2000) as shown in Hinz et al. (1999, figure 14), Planke et al. (2000, figure 9) and Sapin et al. (2021, figure 6). In this paper we use the term SDRs to denote the general observation of volcanic sea-ward dipping reflectors not only applied to those formed in a sub-aerial environment but also to those formed in deep water."

**Comment 4**

In the application of Warner's 10s rule in figure 9 we have added that we used a basement seismic velocity of 6.5 km/s. The updated text reads:

"Warner (1987) observed that the Moho TWTT on marine deep long-offset seismic data was consistently at about 10 s TWTT for thermally equilibrated lithosphere and was remarkably constant (flat) in time irrespective of the complexity of the geology above including sediment thickness variation. Invoking Warner's 10 s rule for the Moho TWTT for thermally equilibrated lithosphere allows the cross-sections shown in Fig. 9c and Fig. 9d to be converted into the time domain as shown in Fig. 9e and Fig. 9f. To do this basement thickness is converted to interval TWTT (using a basement velocity of 6.5 km/s) and subtracted from 10s to give the TWTT of top basement; this estimate of TWTT of top basement is therefore independent of the interval TWTT of bathymetry and post-rift sediments. In the time domain, first volcanics are predicted to occur at ~ 1.5s TWTT for the magma-rich model, while for the magma-normal magmatic addition model first volcanics occur at ~ 5.5 s. These model predictions are consistent with the observed TWTT of first proximal volcanics between 1.2 to 2.2 s for the Torres High Profile S1 and 4.2 to 6.5 s for the Rio Grande Cone Profile S3 shown in Fig. 8. It should be noted that the model prediction assumes a fully equilibrated lithosphere thermal structure while the Pelotas margin with Early Cretaceous breakup age is not yet fully re-equilibrated."

**Comment 5**

We have modified the summary of TWTT of first volcanics as explained in our response to comment 2. The 6[th] bullet point now reads:

• In the time domain, a magma-rich margin, with sub-aerial SDR flows, shows first volcanics between 1.2 to 2.2 s TWTT while a "normal" magmatic margin has first volcanics between 4.2 and 6.5 TWTT.

**Comment 6**

We have added text at the end of section 5.3 as suggested to expand on the usefulness of using TWTT of fist volcanics to distinguish magma-rich margins from margins with normal magmatic addition. The text reads:

"A common classification of rifted margins is whether a margin is magma-rich, magma-normal or magma-poor. An obvious approach to distinguishing a magma-rich from a magma-normal margin, or a margin's position in between these two end members, might appear to be through measurement of the volume of magmatic addition. However the problem arises, as highlighted by Tugend et al. (2020), that in practice thinned continental crust at a rifted margin cannot reliably be distinguished from volcanic extrusives above it and magmatic intrusives (underplate) below for reasons explained by Karner et al. (2021). As a consequence it is not possible to accurately measure the volume of magmatic addition at a rifted margin. Tugend et al. (2019) and Chenin et al (2024) also show that the relative timing of decompression melting with respect to crustal thinning may be as important as magmatic volume in generating a magma-rich margin. As shown above, the TWTT of first proximal volcanics may represent a practical and efficient method of distinguishing a magma-rich form a magma-normal margin, or for placing a margin in between these two end-members."